



# Historic global biomass burning emissions based on merging satellite observations with proxies and fire models (1750-2015)

Margreet J.E. van Marle[1], Silvia Kloster[2], Brian I. Magi[3] Jennifer R. Marlon[4], Anne-Laure Daniau[5], Robert D. Field[6], Almut Arneth[7], Matthew Forrest[8], Stijn Hantson[7], Natalie M. Kehrwald[9], Wolfgang Knorr[10], Gitta Lasslop[2], Fang Li[11], Stéphane Mangeon[12], Chao Yue[13], Johannes W. Kaiser[14], Guido R. van der Werf[1]

[1] Faculty of Earth and Life Sciences, Vrije Universiteit Amsterdam, Amsterdam, the Netherlands
[2] Max Planck Institute for Meteorology, Bundesstraße 53, 20146 Hamburg, Germany
[3] Department of Geography and Earth Sciences, University of North Carolina at Charlotte, Charlotte, North Carolina, USA
[4] School of Forestry and Environmental Studies, Yale University, New Haven, USA
[5] Environnements et Paléoenvironnements Océaniques et Continentaux, UMR EPOC 5805 CNRS, University of Bordeaux, 33615 Pessac, France
[6] NASA Goddard Institute for Space Studies, New York, NY, USA
[7] Karlsruhe Institute of Technology, Institute of Meteorology and Climate research, Atmospheric Environmental Research, 82467 Garmisch-Partenkirchen, Germany
[8] Senckenberg Biodiversity and Climate Research Institute (BiK-F), Senckenberganlage 25, 60325 Frankfurt am Main, Germany
[9] Geosciences and Environmental Change Science Center, U.S. Geological Survey, Lakewood, Colorado, USA
[10] Department of Physical Geography and Ecosystem Science, Lund University, 22362 Lund, Sweden
[11] International Center for Climate and Environmental Sciences, Institute of Atmospheric Physics, Chinese Academy of Sciences, Beijing, China
[12] Department of Physics, Imperial College London, London, United Kingdom
[13] Laboratoire des Sciences du Climate et de l'Environnement, LSCE/IPSL, CEA-CNRS-UVSQ, Universite Paris-Saclay, F-91198 Gif-sur-Yvette, France.
[14] Max Planck Institute for Chemistry, Mainz, Germany

*Correspondence to*: Guido van der Werf (guido.vander.werf@vu.nl)





## Abstract

Fires have influenced atmospheric composition and climate since the rise of vascular plants, and satellite data has shown the overall global extent of fires. Our knowledge of historic fire emissions has progressively improved over the past decades due mostly to the development of new proxies and the improvement of fire models. Currently there is a suite of proxies including sedimentary charcoal records, measurements of fire-emitted trace gases and black carbon stored in ice and firn, and visibility observations. These proxies provide opportunities to extrapolate emissions estimates based on satellite data starting in 1997 back in time, but each proxy has strengths and weaknesses regarding, for example, the spatial and temporal extents over which they are representative. We developed a new historic biomass burning emissions dataset starting in 1750 that merges the satellite record with several existing proxies, and uses the average of six models from the Fire Model Intercomparison Project (FireMIP) protocol to estimate emissions when the available proxies had limited coverage. According to our approach, global biomass burning emissions were relatively constant with 10-year averages varying between 1.8 and 2.3 Pg C year$^{-1}$. Carbon emissions increased only slightly over the full time period and peaked during the 1990s after which they decreased gradually. There is substantial uncertainty in these estimates and patterns varied depending on choices regarding data representation, especially on regional scales. The observed pattern in fire carbon emissions is for a large part driven by African fires, which accounted for 58% of global fire carbon emissions. African fire emissions declined since about 1950 due to conversion of savanna to cropland, and this decrease is partially compensated for by increasing emissions in deforestation zones of South America and Asia. These global fire emissions estimates are mostly suited for global analyses and will be used in the IPCC CMIP simulations.



## 1 Introduction

Fire is one of the most important disturbance agents in terrestrial ecosystems on a global scale, occurring in all major biomes of the world, and emitting roughly 2-3 Pg C year$^{-1}$ mostly in the form of $CO_2$ but also substantial amounts of reduced species and aerosols (Andreae and Merlet, 2001; van der

Werf et al., 2010). Biomass burning activity generally has a strong seasonal cycle and responds to interannual variability and trends in plant productivity, land use, and droughts as well as other climatic factors. Droughts tend to increase fire activity in areas with abundant fuel build-up and decrease fire activity in arid regions (van der Werf et al., 2008). Interactions between climate, humans, and fire are complex and vary both in time and space (Archibald et al., 2009; Bowman et al., 2011). For example,

tropical rainforests in their natural state rarely burn. This is a consequence of moist conditions underneath the canopy and a lack of dry lightning ignitions (Cochrane, 2003). Humans have changed the landscape though using fire for agricultural purposes in tropical forest. Land-use changes, such as logging and forest fragmentation, increased the forest flammability and number of successful lightning-caused ignitions (Aragão and Shimabukuro, 2010; Cochrane and Laurance, 2008; Fearnside, 2005).

Ignitions due to humans have also increased in boreal Asia (Mollicone et al., 2006). However, in many regions humans also suppress fires, both direct via fire fighting and indirect by altering the fire seasonality and by modifying fuel build-up through grazing and prescribed burning (Kochi et al., 2010; Le Page et al., 2010; Rabin et al., 2015).

Our knowledge about how these factors have influenced fire emissions over the past centuries or

millennia has progressively improved over the past decades leading to new biomass-burning emission inventories (Granier et al., 2011). Dentener et al. (2006) reconstructed fire emissions for the year 1750 by scaling GFED fire emissions before the satellite era with population derived from the Hundred Year database for Integrated Environmental Assessments (HYDE, Goldewijk, 2001), assuming that emissions related to deforestation fires were linearly related to population. For other land-surfaces only

60% of the emissions were scaled by population, the remaining 40% remained constant assuming that these fires were natural. For high-northern latitudes the fire emissions were doubled in 1750 to account for present day fire suppression. Other approaches for global fire estimates were often based on the burned area dataset by Mouillot and Field (2005), which consists of gridded data from 1900 onwards,



combining the Along Track Scanning Radiometers (ATSR) observations with historic literature-based results (land use practices, qualitative reports, and country statistics), and tree ring records. This inventory was used to construct the Global Inventory for Chemistry-Climate studies (GICC) dataset (Mieville et al., 2010). GICC provides estimates of biomass burning emissions over the 20$^{th}$ century and

emissions mimicked the patterns in burned area with a decrease over the beginning of the 20$^{th}$ century followed by relatively constant emissions until emissions increased rapidly from the 1980s to 2005. The Reanalysis of the Tropospheric chemical composition (RETRO) inventory estimates global wildfire emissions over 1960 to 2000 with a regional approach by collecting and combining literature reviews with different satellite datasets, and a numerical model with a semi physical approach to estimate fire

spread and fire occurrence. Over 1960 to 2000 RETRO-based fire emissions showed a global significant increase as a result of an increase in tropical forest and peat soil burning (Schultz et al., 2008). The biomass burning emissions dataset used in the Intergovernmental Panel on Climate Change (IPCC) Fifth Assessment report (AR5) estimated biomass burning emissions from 1850 through 2000 (Lamarque et al., 2010) using a combination of GICC for 1900-1950, the RETRO inventory for the

1960-1997 period, and the satellite-based Global Fire Emissions Database (GFED) version 2 for 1997 through 2000 (van der Werf et al., 2006). For the 1850-1900 time period biomass burning emissions were held constant because no additional data were available (Mouillot and Field, 2005). The reconstructed global signal indicated that fire emissions were relatively stable until the 1920s. They then decreased until 1950s, after which they increased until the end of the dataset in 2000 (Lamarque et

al., 2010).

Besides these estimates based on historic datasets and satellite data, individual fire models can also be used to estimate biomass burning emissions on a global scale (Figure 1). Over the past decades these models have been embedded in dynamic global vegetation models (DGVMs), Earth system models (ESMs) and terrestrial ecosystem models (TEMs), and by this method the feedbacks between fire and

vegetation can be examined (Hantson et al., 2016). These models have been growing in complexity and a large variety of models now exist. To better analyse and evaluate these models the Fire Model Intercomparison Project (FireMIP) was initiated, where models were forced with the same forcing (meteorology, lightning, land-use, population density, atmospheric $CO_2$) datasets (Rabin et al., 2016).



While fire models in general have a global focus, they often do not include anthropogenic fires used in the deforestation process. However, another data source is available to estimate these fires: the country-level estimates of deforestation and afforestation provided by the United Nations' Food and Agricultural Organization's (FAO) Forest Resource Assessment (FRA) (Food and Agriculture

Organization of the United Nations, 2012). These estimates are subsequently used in a bookkeeping model to calculate carbon emissions (Houghton, 2003).

All these emission inventories rely on different datasets and different assumptions. The most consistent estimates of fire patterns are based on satellite-observed burned area or active fires. These usually have a high temporal resolution and are available globally (Figure 1). These satellite observations are used in

combination with a biogeochemical model to estimate fuel loads and calculate emissions in the GFED (van der Werf et al., 2010) or using fire-emitted energy scaled to GFED in the Global Fire Assimilation System (GFAS, Kaiser et al., 2012). Unfortunately these datasets only cover about 2 decades, i.e. since 1997 for GFED and shorter for other datasets, including those based on atmospheric observations of fire-emitted species which can be used to infer emissions when combined with an atmospheric transport

model (Edwards et al., 2006; Huijnen et al., 2016; Krol et al., 2013).

Proxy records cover longer time scales, of which the charcoal record is probably the most extensively explored (Daniau et al., 2013; Marlon et al., 2013; Power et al., 2008). Charcoal records can be used for reconstructing fire emissions on a local to regional scale covering time periods of decades to millennia and beyond. Regional and global scale analyses have been done compositing multiple records within a

region or globally. The Global Charcoal Database (GCDv3) consists of 736 charcoal records globally, with most samples taken in North America, Europe, Patagonia and South-East Australia (Marlon et al., 2016). Ice cores are another widely used proxy for retrieving information about fire history on decadal to longer time scales and are representative for regional to continental scales. The most commonly used proxy records are based on concentrations and isotopic signatures of CO and $CH_4$ (Ferretti et al., 2005;

Wang et al., 2010, 2012).

These charcoal records and isotopic ratios of CO and $CH_4$ have also been used to reconstruct the fire signal, most often focusing on the last centuries or millennia. The charcoal records suggest that despite close links between fires and humans, pre-industrial fires were not necessary lower than present-day.




The charcoal record also shows that fire has been continuously present in both populated and unpopulated areas since the last glacial maximum (Power et al., 2008) with no major change in regional fire regime coinciding with the arrival of modern humans in Europe or Australia (Daniau et al., 2010; Mooney et al., 2011). The charcoal-based global analysis of Marlon et al. (2008) indicated a gradual

decrease from 1AD until 1750AD, possibly as a result of a global cooling trend. Over the $16^{th}$ and $17^{th}$ century the lowest emissions were observed, coinciding with the climate-driven little ice age (LIA). Based on $CH_4$ concentrations and its isotopic ratio Ferretti et al. (2005) have hypothesized that this decrease of human-driven fires in the South American tropics resulted from decreased human-caused burning after the arrival of Europeans and the introduction of diseases in the tropics, but decreased

burning is evident in both the Americas and globally (Power et al., 2013), and thus is better explained by widespread cooling during the LIA. Later-on biomass burning emissions increased and peaked in the late $19^{th}$ century. This peak was also seen in an Antarctic ice core record of CO concentrations and its isotopic ratio (Wang et al., 2010). Observations of $CH_4$ concentrations and its isotopic ratio also indicated an increase, however this increase continues until present, without a peak in the $19^{th}$ century

(Ferretti et al., 2005). This pattern is also observed in firn air samples in both the Northern (Wang et al., 2012) and Southern Hemisphere (Assonov et al., 2007).

Although isotopic ratios of CO and $CH_4$, and charcoal thus show similar features there are key differences. These differences over the past decade could be the result of different lifetimes of CO (two months, providing more regional information) and $CH_4$ (about a decade, providing information on a

global scale), but also because of the distribution of the charcoal datasets, which is denser in temperate regions than in the tropics. Besides this disagreement in trends over the past decade, the amplitude seen in the only known CO record is much larger than in the $CH_4$ records and is difficult to explain with our current understanding of fire emissions (van der Werf et al., 2013).

Besides the most often used charcoal and ice core CO and $CH_4$ records, other proxies have recently

been employed to reconstruct fire histories. Field et al. (2009) used horizontal visibility data as observed by weather stations to show how increases in fire emissions were linked to transmigration in Indonesia. Their record started in 1960. A similar approach was used by van Marle et al. (2016) but focused on the Amazon where a similar pattern was found.





Finally, ice core and firn records of levoglucosan, a specific biomass burning marker, have enabled the reconstruction of boreal fire emissions for the past two millennia (Kehrwald et al., 2012a; Zennaro et al., 2014) and black carbon concentrations taken from ice cores have been used to reconstruct fossil fuel and biomass burning emissions from boreal sources over the past 220 years (McConnell et al., 2007).

Excess ammonium in ice cores has been used as a fire proxy on very long time scales (Fischer et al., 2015), and in rare cases multi-proxy fire reconstruction have also been developed from ice cores (Eichler et al., 2011; Legrand et al., 2016).

To reconstruct fire emissions, there is thus a wide range of information available, each with strengths and weaknesses. The observation-based visibility data provides annual data but is only available for
deforestation regions and extends the satellite-record only by a few decades. The charcoal data provides a much longer record and is most useful in temperate and boreal regions where data density is highest, but the signals are unitless and it is unknown what the signal exactly represents. Focusing on the strengths of these different data sources may provide a more complete history of fire on Earth (Kaiser and Keywood, 2015; Kehrwald et al., 2016). We have reconstructed global fire emissions since 1750
using observation-based data streams (fire emissions based on satellite data for the 1997 onwards period, charcoal datasets in temperate and boreal regions, and visibility-records from weather stations in deforestation zones of South America and Indonesia) and multi-model mean emission estimates from FireMIP when no observations were available, and anchored them to satellite-based fire emissions.





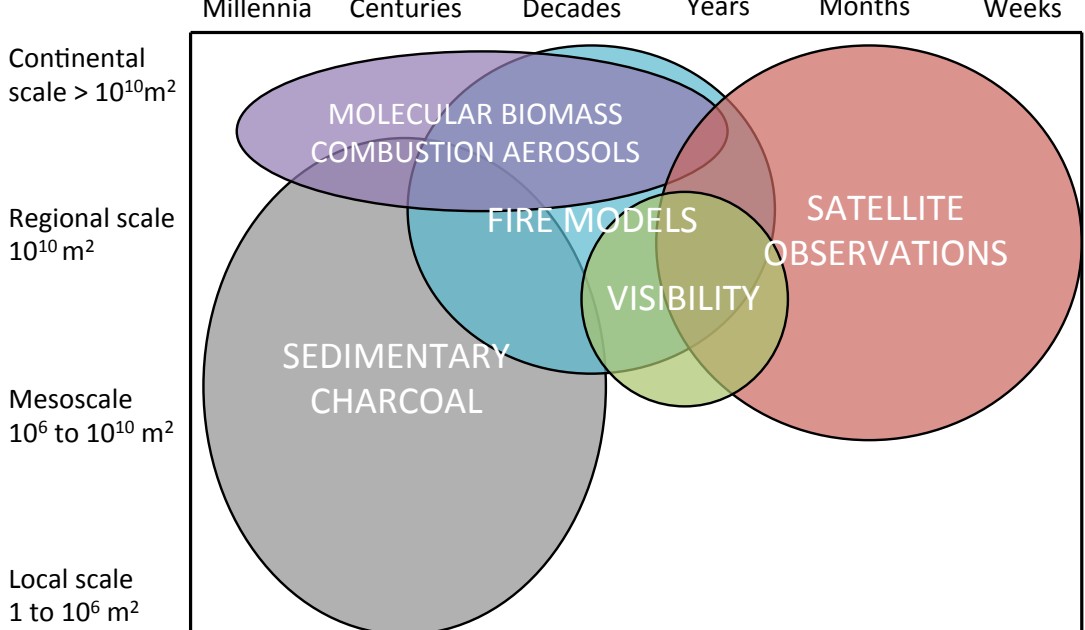

**Figure 1. Spatial and temporal resolution of various data streams available to estimate fire emissions. Adapted from Kehrwald et al. (2016).**



## 2 Datasets and Methods

To leverage the specific strengths of the various proxies, we divided the globe into the 14 regions used within GFED, which feature relatively homogeneous fire seasons and characteristics, but further sub-divided some of these regions to allow input from additional datasets for a total of 17 regions (Figure
5   2).

For these 17 regions, we combined the satellite-based emissions from GFED (version 4s) for 1997 to 2015 with either proxies (when available), or fire models to calculate the fire history since 1750 (Figure 3). We used visibility observations from the World Meteorological Organization (WMO) stations in the Arc of Deforestation (ARCD) and Equatorial Asia (EQAS). Dimensionless charcoal records were
10   scaled to the range of the fire models and were used for Europe (EURO) and North America, where boreal and temperate North America was split in an eastern (BONA-E, TENA-E) and western (BONA-W, TENA-W) part. For all other regions no proxy observations were available and we used the median of fire model outputs anchored to GFED4s to extrapolate back to 1750. Both proxies and models were only used to reconstruct annual regional totals, these were distributed over the 0.25°×0.25° grid and
15   months based on the GFED4s climatological patterns (1997-2015). In the next sections we describe the datasets and methods in more detail.

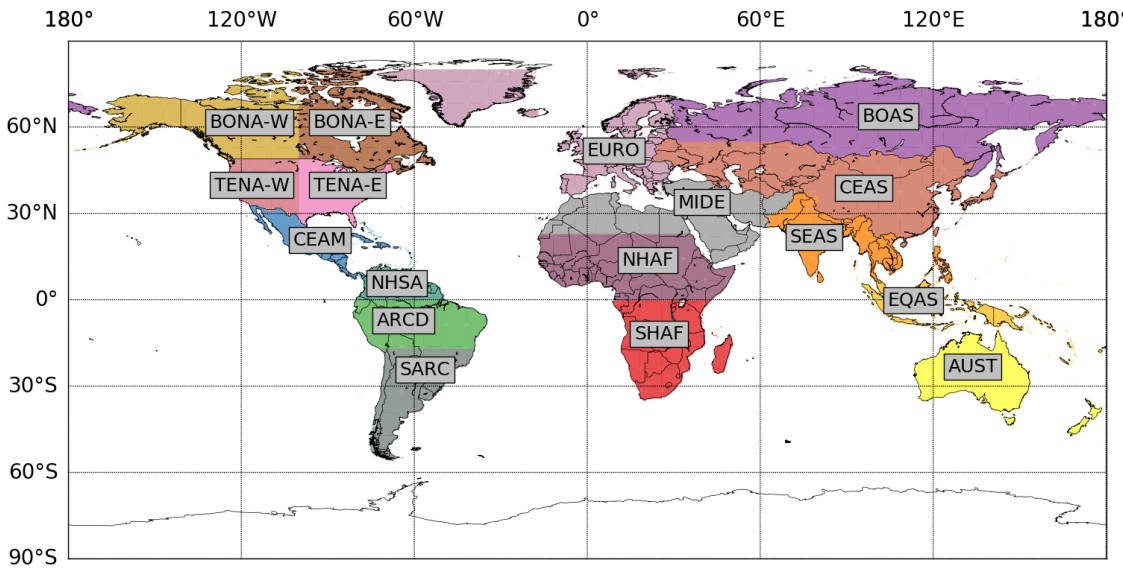

**Figure 2: The 17 basis-regions used to reconstruct fire emissions, abbreviations are explained in Table 1.**



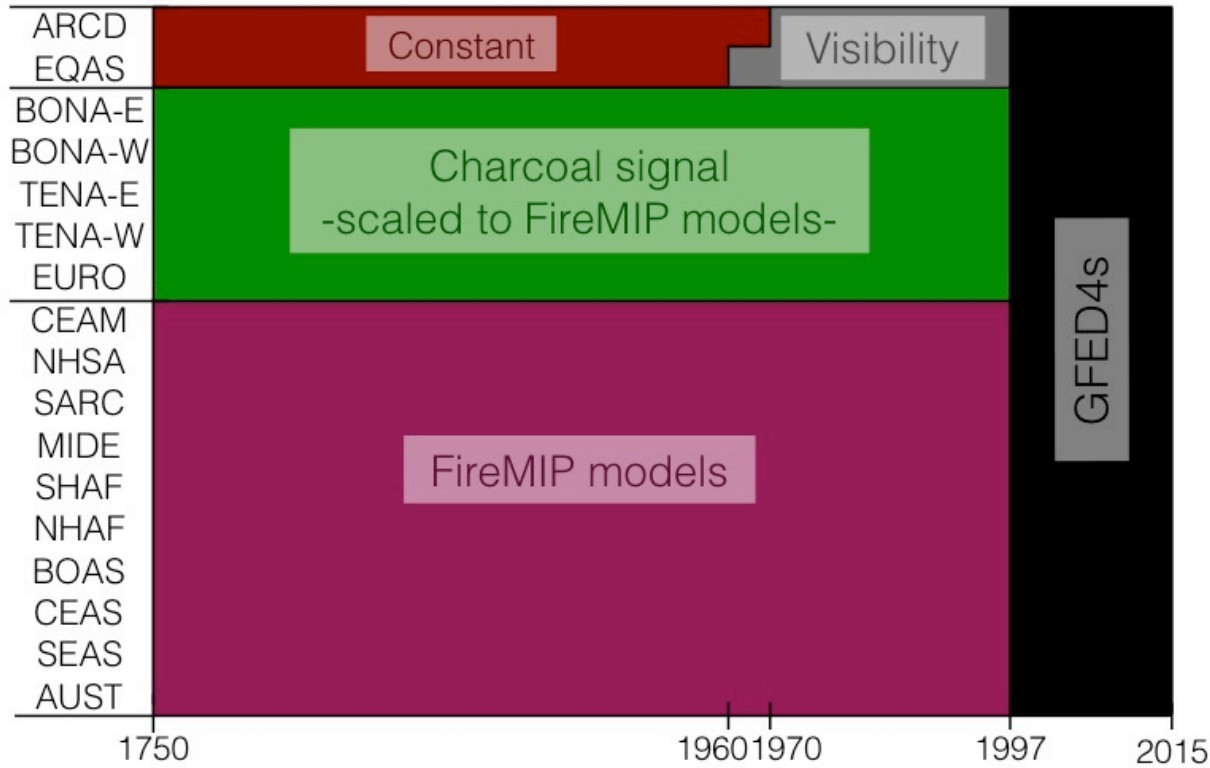

**Figure 3: The data sources used for each region.**

## 2.1 Global Fire Emissions Database (GFED)

5 We used the Global Fire Emissions Database version 4 with small fires (GFED4s) for 1997-2015 and as anchor point for all proxies and model results. In GFED, satellite derived burned area is used as a key input dataset in a revised version of the Carnegie Ames Stanford Approach (CASA) biogeochemical model (Potter et al., 1993). The burned area estimates from 2000 onwards are from the Moderate Resolution Imaging Spectroradiometer (MODIS) MCD64A1 500-meter burned area maps aggregated to

10 0.25°×0.25° spatial resolution and a monthly time step (Giglio et al., 2013). These estimates are 'boosted' using a revised version of the small fire estimates of Randerson et al. (2012) which are based on overlaying mapped burned area and active fires. Finally, the burned area estimates are used in combination with active fire detection from the Visible and Infrared Scanner (VIRS) and the Along-



Track Scanning Radiometer (ATSR) sensors to extend this time series back to 1997 (van der Werf et al., 2017).

In CASA, the burned area estimates are then converted to carbon emissions using modelled fuel consumption. Fuel consumption depends on the amount of flammable biomass and combustion completeness, and is calculated in the model as a function of satellite-derived plant productivity, fractional tree cover estimates, and meteorological datasets including solar insolation and moisture levels (van der Werf et al., 2010, 2017). The fuel consumption parameterization in the model was tuned to match observations compiled by van Leeuwen et al. (2014). As a final step, these carbon emission estimates are converted to trace gas and aerosol emissions using emission factors based mostly on the compilation of Akagi et al. (2011) with lumping described in http://www.falw.vu/~gwerf/GFED/GFED4/ancill/.

## 2.2 Fire models

The global fire models used here were scaled to GFED4s and used in regions where no proxy data were available, and also to set upper and lower bounds for those regions where charcoal observations were used (Figure 3). The latter will be described in more detail in Section 2.4.

There are generally two types of fire models embedded in global dynamic vegetation models (DGVM's). In 'process-based models' fires are simulated from a mechanistic point of view, with fire number and size being separately simulated to derived burned area. Fire size simulation often takes into account fire propagation and duration under given weather conditions and are also influenced by fuel state, human suppression and economic conditions. In contrast 'empirical models' are based on statistical relationships between, amongst others, climate and population density with usually burned area (Hantson et al., 2016). Models are developed with different complexity and some models combine both empirical and process-based approaches. We used all five model outputs available at the time (May 14, 2016) within FireMIP, which covers the 1750-2013 time period, as well as one model that did not participiate in FireMIP, the SIMFIRE-GDP model. These six models are described in more detail below. FireMIP has a main goal to evaluate fire models with benchmark datasets to understand differences between models and improve the representation of fires in DGVMs. The models within

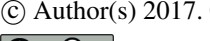



FireMIP used identical forcing datasets with prescribed climate, $CO_2$, lightning, population density, and land-use from 1700 to 2013 (Rabin et al., 2016).

We aggregated carbon emissions for each model (mod) and region (reg, Figure 2) to an annual time step (yr). These estimates were then scaled for each individual model to the regional GFED fire emissions
for the overlapping 1997-2003 time period:

$$FireMIP_{scaled}(reg, yr, mod) = \frac{FireMIP_{data}(reg, yr, mod)}{FireMIP_{1997:2003}(reg, mod)} \times \overline{GFED}_{1997:2003}(reg) \qquad (1)$$

where $FireMIP_{1997:2003}(reg, mod)$ is the average regional emission estimate for 1997-2003, and
$FireMIP_{scaled}(reg, yr, mod)$ the scaled regional model output on an annual time step. In regions where no proxy information was available and where we therefore only used model output (Figure 3), fire emissions before 1997 were based on the median of the 6 FireMIP$_{scaled}$ time series. We used the average over 1997 to 2003 when combining the various data streams to minimize the impact of interannual variability on our rescaled FireMIP emissions. Below we will describe the models we used here,
followed by a description of other datasets used and how the various pieces of information were merged to regional time series of emissions for the 1750-2015 period.

### 2.2.1 CLM

The fire module used in the National Center for Atmospheric Research (NCAR) Community Land Model was version 4.5 (CLM4.5). The fire module embedded in CLM consists of four components:
non-peat fires outside croplands and tropical closed forests, agricultural fires in croplands, deforestation fires in the tropical closed forests, and peat fires. The first component is process-based, in which burned area is simulated as the product of fire counts and average fire size and regulated by weather and climate, vegetation characteristics, and human activities (Li et al., 2012, 2013) Anthropogenic ignitions and fire suppression are functions of population density and gross domestic product (GDP) per capita.
The other three components are empirical (Kloster et al., 2012; Li et al., 2013). Burned area depends on socioeconomic factors, prescribed fire timing, and fuel load for agricultural fires, climate and deforestation rate for tropical deforestation fires, and climate and area fraction of peat exposed to air for

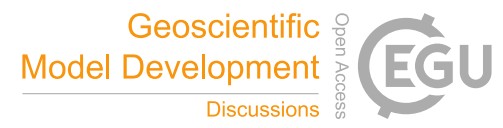

peat fires. The simulated burned area is then converted to fire carbon emissions based on simulated biomass and plant functional type (PFT)-dependent combustion completeness factors for leaves, stems, roots and litter (Li et al., 2012, 2014)

### 2.2.2 INFERNO

The INteractive Fire and Emissions algorithm for Natural envirOnments (INFERNO, Mangeon et al., 2016) model was developed to incorporate a fire parameterization into the Joint UK Land Environment Simulator (JULES) and eventually into an ESM. INFERNO is a reduced-complexity empirical global fire model that builds on the parameterization for fire occurrence from Pechony and Shindell (2009). It estimates burned area and emissions for each of the PFTs used in JULES. Fuel flammability is

determined at each time step (using temperature, relative humidity, fuel density, precipitation and soil moisture). Ignitions are calculated using population density and cloud-to-ground lightning. Burned area is derived from fire occurrence using a fixed average burned area for different vegetation: 0.6, 1.4 and 1.2 km$^2$ for trees, grasses and shrubs respectively. Carbon emissions are then estimated using biomass densities from JULES area and combustion completeness, which scales linearly with soil moisture for

leaves (between 0.8 and 1) and stems (between 0 and 0.4).

### 2.2.3 JSBACH, LPJ-GUESS, ORCHIDEE

In three of the DGVMs the SPread and InTensity of FIRE (SPITFIRE) model serves as the fire module (Thonicke et al., 2010). SPITFIRE is a process-based global fire model and a further development of the Reg-FIRM approach (Venevsky et al., 2002), but uses a more complete set of physical

representations of spread and fire intensity. Precipitation, daily temperature, wind speed, soil moisture, carbon content of the vegetation and litter pools, and the vegetation distribution are used as input for SPITFIRE to calculate rate of spread, fire duration and intensity. Based on the calculated burned fraction and post-fire mortality of trees, carbon emissions are calculated and redistributed over carbon pools. SPITFIRE includes a dynamic scheme for combustion completeness and depends on fire

characteristics and the moisture content of different fuel classes (Lenihan et al., 1998; Thonicke et al., 2010). SPITFIRE was originally developed for the Lund-Potsdam-Jena (LPJ) vegetation model and is





modified for the use within the Jena Scheme for Biosphere-Atmosphere Coupling in Hamburg (JSBACH), the Lund-Potsdam-Jena General Ecosystem Simulator (LPJ-GUESS) and the ORganizing Carbon and Hydrology In Dynamic EcosystEms (ORCHIDEE).

The JSBACH land surface model (Brovkin et al., 2013; Reick et al., 2013) is the land component of the

Max-Planck-Institute Earth System Model (MPI-ESM) (Giorgetta et al., 2013). Differences with the original SPITFIRE model are that the vegetation distribution is prescribed and includes two shrub PFTs. The relation between rate of spread and wind speed was modified (Lasslop et al., 2014). Human ignitions and a coefficient related to the drying of fuels were adjusted. Furthermore the combustion completeness values were updated to better mimic field observations (van Leeuwen et al., 2014).

In contrast to the original LPJ model, the LPJ-GUESS vegetation model (Smith et al., 2001) follows a 'gap-model' approach and simulates stochastic establishment and mortality of trees in multiple replicate plots (referred to as patches) for each modelled locality. This allows trees of different sizes and ages to co-exist and thus provides more detailed representation of vegetation structure and dynamics. Therefore the original SPITFIRE model was integrated into LPJ-GUESS (Smith et al., 2001) and was adapted to

take advantage of these features. Most importantly the fire characteristics are calculated separately for each patch and the burned area for a patch is interpreted as the probability of a particular patch burning, rather than as a fraction of the locality which burns (Lehsten et al., 2009). As a further consequence of the more detailed vegetation structure, the size dependent mortality functions in SPITFIRE have a more realistic impact, whereby small trees have a relatively higher probability of being killed by fires than

large trees. For the FireMIP simulations used here further improvements were made; the calculation of human ignitions was recalibrated and post-fire mortality parameters were updated.

For the global vegetation model ORCHIDEE (Krinner et al., 2005), SPITFIRE was adjusted and incorporated by Yue et al. (2014, 2015). Most equations from the original SPITFIRE model were implemented and run parallel to the STOMATE sub-module which simulated vegetation carbon cycle

processes in ORCHIDEE. Minor modifications are made by Yue et al. (2014, 2015) and include updated combustion completeness values based on field measurements (van Leeuwen et al., 2014).



### 2.2.4 SIMFIRE-GDP

We used the stand-alone semi-empirical simple fire model (SIMFIRE) coupled to LPJ-GUESS (Knorr et al., 2016), after optimising SIMFIRE according to Knorr et al. (2014), albeit with a modified semi-empirical function (see Appendix A). SIMFIRE is an empirical global fire model, where burned area

estimates are based on human drivers (only population density in the original version) as well as climate and remotely sensed vegetation factors (the fraction of absorbed photosynthetically active radiation (FAPAR, Gobron et al., 2010)) as environmental drivers. The version used here relies additionally on large-region averages of per capita gross domestic product (GDP) in combination with human population density as statistical drivers for land use impacts on burned area. Simulations with the

original coupled LPJ-GUESS-SIMFIRE global dynamic vegetation–wildfire model (Knorr et al., 2016) revealed that over the 20th century population density was the main driver of wildfire emissions, whereas climate factors only had a small influence. Therefore, prior to 1900, only GDP and population density are used to re-scale emissions computed by LPJ-GUESS-SIMFIRE for the early 20th century, resulting in no climate-driven interannual variability.

### 2.3 Visibility-based fire emission estimates

Fire-emitted aerosols lower visibility, and in the frequently burning regions of EQAS (Field et al., 2009) and ARCD (van Marle et al., 2017) visibility observations can be used as a proxy for fire emissions given the reasonable agreement between fire emission estimates from GFED and visibility observations for the overlapping 1997-2015 period. The visibility observations are taken from weather station

records from the NOAA National Centers for Environmental Information (NCEI) Integrated Surface Database (ISD). For EQAS data are available from 1950 onwards and for ARCD data are available starting in 1973. Fire emissions in these regions have increased over time related to migration of humans accompanied with deforestation (Field et al., 2009; van Marle et al., 2017).

We replaced the visibility-based emissions from 1997 through 2015 with the estimates based on

GFED4s (Figure 4). To extend this combined time-series to years with no visibility-observations, we kept the emissions constant at the lowest decadal average. This approach is based on the assumption that fires do not occur naturally in these regions and that fires here are strongly linked to population



density (Fearnside, 2005; Field et al., 2009). This strong link is also seen when comparing HYDE 3.1 population density (Klein Goldewijk et al., 2011) and visibility-based fire emissions with an $r^2$ of 0.67 in the Arc of Deforestation and an $r^2$ of 0.84 in Equatorial Asia (both with $p<0.05$) over 1750 to 2000.

**Figure 4: GFED4s-based fire carbon emissions, visibility-based fire carbon emissions and constant carbon emissions for ARCD (top) and EQAS (bottom).**

## 2.4 Global Charcoal Database

The Global Charcoal Database version 3 (GCDv3) is the most recent version of the GCD (Marlon et al., 2016) and includes 736 charcoal records. The records are distributed over 5 continents with the majority of sites having one record. The sites are not distributed evenly over the globe: many sites (326) are located in Northern America and Europe. Records may lack data from the most recent 250 or so years





(the near-surface sediment), which further restricts the charcoal analysis (see Figure B1 with locations of charcoal sites and regions in Appendix B). While all our regions have charcoal records, data density is highest in temperate and boreal regions (in total for 5 regions, Figure 3). The charcoal records were converted to unitless time series with a decadal time step using methods detailed in Power et al. (2010),

which is then scaled to the output of the modelled data described under 2.2 following Eq. 2:

$$CC_{FireMIP}(reg, yr) = \overline{FMIP}_{25th}(reg) + \left(CC_{norm}(reg, yr) * (\overline{FMIP}_{75th}(reg) - \overline{FMIP}_{25th}(reg))\right) \quad (2)$$

where the normalized charcoal signal ($CC_{norm}$) is the unitless charcoal influx Z-score on a decadal time

step normalized per region and year to values between 0 and 1 following the approach described in (Power et al., 2010). Here a base period of -60 to 200 cal yr BP (1750-2010 AD) was used to obtain a common mean and variance for all sites. The composite curves per region were obtained using a locally weighted regression with a window (half) width of 10 yrs. $\overline{FMIP}_{25th}$ and $\overline{FMIP}_{75th}$ are the average regional 25$^{th}$ and 75$^{th}$ percentiles based on the output of the 6 FireMIP models for 1750-2000. We used

the 25$^{th}$ to 75$^{th}$ percentiles, so outliers did not influence the scaled charcoal signal. To stitch the regional charcoal signal to the GFED period the charcoal signal adjusted to the FireMIP model output ($CC_{FireMIP}$) is scaled to the average regional GFED carbon emissions over 1997 through 2003 ($\overline{GFED}_{1997:2003}$, Eq. 3), similar to scaling the FireMIP models to GFED. This is done in the same fashion as when scaling plain model results to GFED, thus averaging out the large interannual

variability in fire emissions.

$$CC_{scaled}(reg, yr) = \frac{CC_{FireMIP}(reg, yr)}{CC_{FireMIP}(reg, 2000)} * \overline{GFED}_{1997:2003}(reg) \quad (3)$$

### 2.5 Breakdown of regional fire emissions

The annual regional fire emissions over 1750 to 2015 were distributed over the 0.25°×0.25° grid cells

based on the GFED4s climatology (1997-2015). We thus assumed that within each region the spatial and monthly patterns did not change over time. When fire modules are embedded in climate models they may be in a better position to include some spatial and temporal variability based on simulated



weather. The contributions of emissions related to deforestation fires, fires in boreal and temperate forests, savanna fires, agricultural waste burning on field, and peatland fires were again based on the GFED climatology. Areas where deforestation and peat fires were important had declining emissions going back in time. Agricultural fires were relatively constant over time as we did not adjust the relative

contribution of these fires due to a lack of information and fire emissions in these regions did not decline as much as in deforestation zones going back in time. This partitioning was used to convert carbon emissions to the different emissions of several species based on the same emission factors as used in GFED (http://www.falw.vu/~gwerf/GFED/GFED4/ancill/). The emissions for BC, $CH_4$, CO, $H_2$, $N_2O$, $NH_3$, NMVOC, $NO_x$, OC, and $SO_2$ were provided. The NMVOC emissions consists of the

sum of $C_2H_6$, $CH_3OH$, $C_2H_5OH$, $C_3H_8$, $C_2H_2$, $C_2H_4$, $C_3H_6$, $C_5H_8$, $C_{10}H_{16}$, $C_7H_8$, $C_6H_6$, $C_8H_{10}$, Toluene_lump, Higher_Alkenes, Higher_Alkanes, $CH_2O$, $C_2H_4O$, $C_3H_6O$, $C_2H_6S$, HCN, HCOOH, $CH_3COOH$, MEK, $CH_3COCHO$, $HOCH_2CHO$ (Akagi et al., 2011).

## 3 Results

### 3.1 Global fire emissions

According to our approach, global biomass burning emissions were relatively stable from 1750 to 2015 (Figure 5). Carbon emissions increased only slightly over the full time period and peaked during the 1990s after which they decreased gradually. Although Africa exhibits a decrease from 1950 onwards, this decline in emissions was compensated for, especially in the 1990s, by increasing emission in deforestation zones (Figure 5). From 1960 onwards the interannual variability increased in our dataset

as a result of more detailed information from the visibility-based (1960 to 1997) and satellite-based (1997-2015) biomass burning emission datasets. The cyclic variability in the first centuries is related to the use of repeating climate variability in FireMIP. While the increase in IAV is thus partly due to changes on underlying data sources, it has also increased in reality because of the increase in deforestation-based emissions which vary more from year to year than other fire emissions sources.

The global trend in fire emissions reflects mostly the patterns in biomass burning emissions from Africa, which contributed more than half (58%) to the global biomass burning emissions from 1750 to



2015 (Figure 6), where southern hemisphere Africa (SHAF) contributed more (33%) than northern hemisphere Africa (NHAF, 25 %). Tropical America (9%) and tropical Asia (equatorial Asia (EQAS) and southeast Asia (SEAS) combined, 14%) are regions substantially influenced by land-use change and contributed most after Africa. These regions are followed by boreal (8%) and temperate (6%) regions, and Australia (5%) (Table 1).

Figure 5: Global biomass burning carbon emissions (1750-2015).





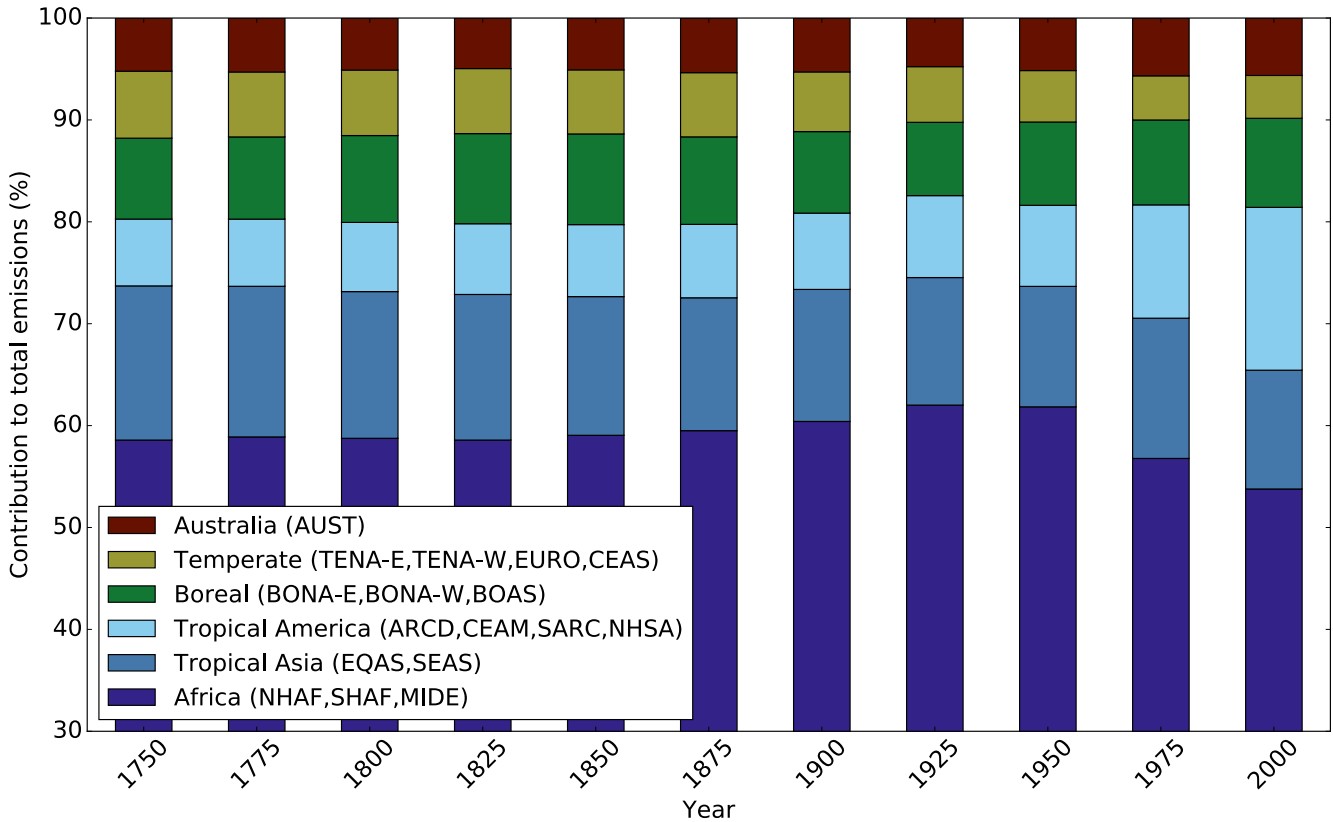

Figure 6: Relative contribution of various regions to global fire emissions. Contributions are calculated as averages over 25 years, except for the 2000-2015 period, which is based on 16 years. Note the contribution starts at 30%.

## 3.2 Regional breakdown of estimates

### 3.2.1 Africa

The multi-model median indicated that Southern Hemisphere Africa (SHAF) had a slight increasing trend from 1750 until ~1950, after which emissions stabilized. Not all models agreed on this: the two models that departed most from the average were SIMFIRE which had a decreasing trend in fire emissions and highest emissions in preindustrial times, and ORCHIDEE showing a stronger increasing trend (**Figure 7**). In Northern Hemisphere Africa (NHAF) emissions were relatively constant from 1750 until the 1950s, after which the emissions decreased, first slightly and from 1997 onwards more steeply, until present-day (**Figure 7**). All models, except CLM, agreed with this decreasing trend. Therefore, the




range in the 25th to 75th percentile was relatively small. The Middle-east (MIDE), including the African Sahara, contributes little (0.2%) to global emissions. These emissions were stable until 1900 after which they decreased, all models agreed on this trend (**Figure 7**).



5   **Figure 7: Fire carbon emissions for African regions. Panels on the left indicate all model outputs scaled to the average GFED values over 1997-2003 for that region. The panels on the right indicate the median of the models in purple (solid line) and the GFED signal in black. The variation between the models is shown in pink (25[th] to 75[th] percentiles) and light pink (total range models).**

## 3.2.2 South America

10   In the Arc of Deforestation (ARCD) biomass burning emissions were based on visibility-observations from weather stations from 1973 to 1997 and GFED4s emissions estimates from 1997 to present (**Figure**



**8**). According to this approach fire emissions were constant with 32 Tg C yr-1 until 1973, after which they stayed relatively low until the first high fire years in 1987 and 1988. After that fire emissions increased rapidly with fire emissions of an average of 280 Tg C yr$^{-1}$ over the 2000s and highest values often coinciding with El Niño years.

Other tropical regions in South America are Central America (CEAM, contributing 2.4%), Northern Hemisphere South America (NHSA contributing 1.4%) and South of the Arc of Deforestation (SARC, contributing 2.7%) (**Figure 8**). In these regions the fire emissions were based on the median of scaled models. The 25$^{th}$ to 75$^{th}$ percentile range was relatively small and for all three regions most models showed a decrease from 1950 to present. In the SIMFIRE model the decrease started around 1900. In

SARC most models showed an increase until the decrease from 1950 onwards.

### 3.2.3 Tropical Asia and Australia

In Equatorial Asia (EQAS) biomass burning emissions were also based on visibility-observations. Here the emissions were kept constant at 26 Tg C yr$^{-1}$ until 1960 based on the average emissions over 1955-1965, when the visibility observations started, after which they increased with large interannual

variability (**Figure 9**). The highest fire year was 1997, followed by 1991, 1994 and 2015, all El Niño years. South East Asia (SEAS) is another tropical Asian region contributing 11.0% to the global budget (**Figure 9**). Here, the models also showed a decreasing trend over time, where SIMFIRE exposed the highest pre-industrial emissions, decreasing strongly from 1950 to present.

Australia (AUST) contributed 5.2% to the global budget and the median value is relatively constant

over time, with only a small sudden jump in the 1970s. The models exhibited a large range in emissions, where CLM presented higher values in 1750 compared to the other models (**Figure 9**).



**Figure 8: Fire carbon emissions for Central and South American regions. Panels on the left indicate all model outputs scaled to the average GFED values over 1997-2003 for that region. The panels on the right indicate the visibility-based fire emissions in grey, the median of the models in purple (solid line) and the GFED signal in black. The variation between the models is shown in pink (25th to 75th percentiles) and light pink (total range models).**





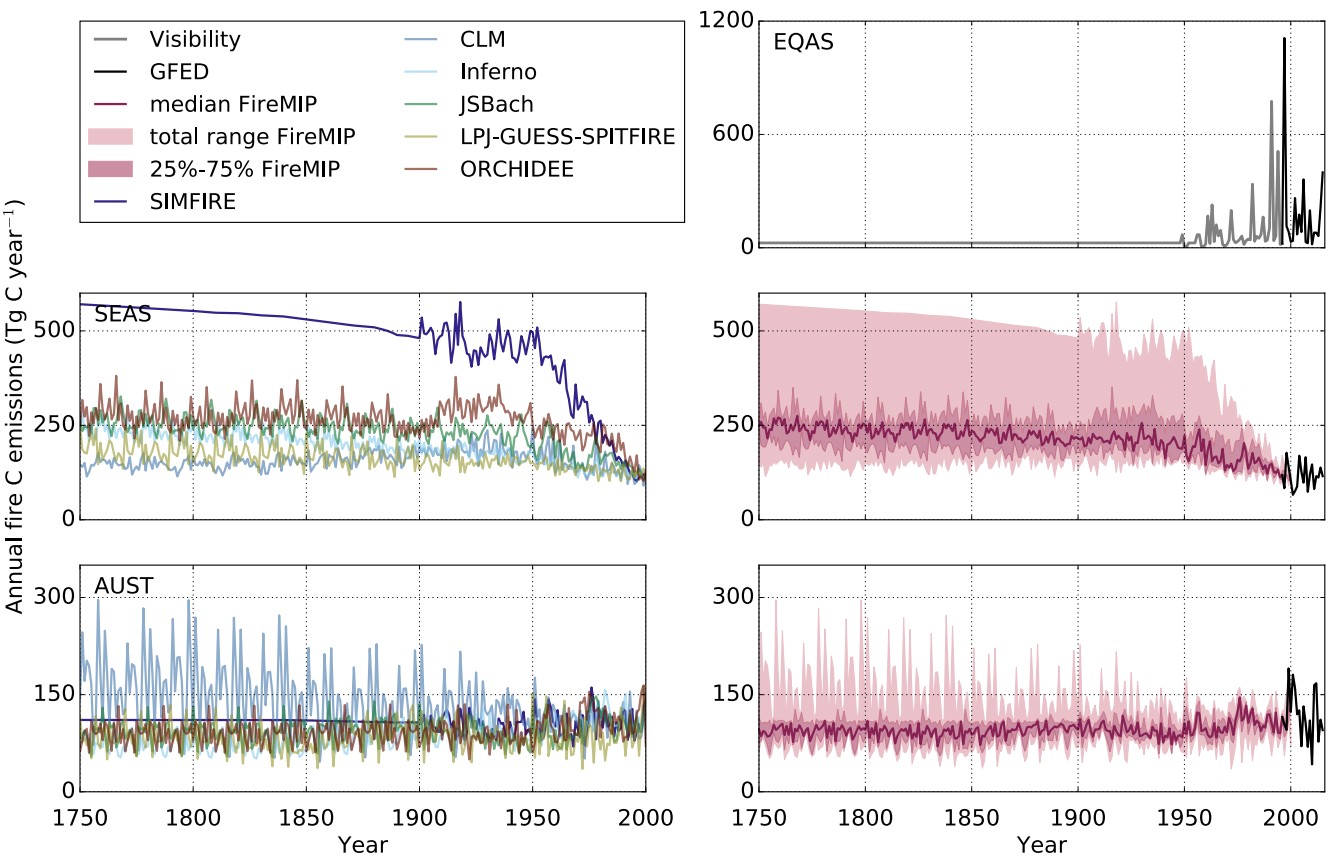

**Figure 9: Fire carbon emissions for Tropical Asian regions and Australia. Panels on the left indicate all model outputs scaled to the average GFED values over 1997-2003 for that region. The panels on the right indicate the visibility-based fire emissions in grey, the median of the models in purple (solid line) and the GFED signal in black. The variation between the models is shown in pink (25$^{th}$ to 75$^{th}$ percentiles) and light pink (total range models).**

### 3.2.4 Boreal regions

In both western boreal North America (BONA-W), contributing 2.2%, and eastern boreal North America (BONA-E), contributing 0.7% to the global fire emissions, the number of charcoal records was relatively dense and used here to represent the regional signal with the upper and lower bounds set by the 75$^{th}$ and 25$^{th}$ percentile of the models (**Figure 10**). According to this approach the levels in biomass burning emissions BONA-W were in 1750 about the same as present-day. After a peak in 1850 fire emissions decreased until 1920 after which biomass burning emissions started to increase until present. Agreement with models was poor; most models showed an increase from 1750 to present, only





JSBACH and SIMFIRE had a relatively stable period from 1750 until 1900, after which emission decreased.

**Figure 10: Fire carbon emissions boreal regions. Panels on the left indicate all model outputs scaled to the average GFED values over 1997-2003 for that region. The panels on the right indicate the charcoal signal in green (solid line), the median of the models in purple (solid line) and the GFED signal in black. The variation between the models is shown in pink (25$^{th}$ to 75$^{th}$ percentiles) and light pink (total range models).**

In eastern boreal North America (BONA-E) the charcoal signal was relatively constant, something most models agreed on. The charcoal signal did have small peaks just before 1800 and 1900 and after a small decrease, emissions started to increase until present.

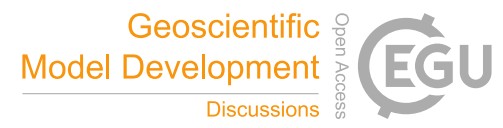

Biomass burning emissions in Boreal Asia (BOAS, contributing 5.4%) were based on the median of the 6 models. The model simulations showed in general less interannual variability than GFED and taking their median decreased the variability even further. Also, the median exhibited no clear in- or decreasing trend, thus the regional signal stayed relatively constant, while the range between the models was relatively large.

### 3.2.5 Temperate regions

The regions western temperate North America (TENA-W), eastern temperate North America (TENA-E), and Europe (EURO) were all based on the charcoal signal (Appendix B) with upper and lower limits based on the models, just like in the boreal regions (**Figure 11**). These three regions combined contributed 1.6% to the global total. The pattern based on the charcoal signal in TENA-W showed a peak in 1850 after which fire emissions decreased until 1920. Afterwards they increased to the present, a pattern similar to the BONA-W trend. The models had a relatively large range, SIMFIRE and CLM exhibited a decrease from 1750 to present, all other models were relatively low until 1850 after which they increased.

The charcoal records in TENA-E indicated relatively constant emissions until 1800, after which emissions increased until a peak in 1900. From 1900 until present-day emissions decreased again. The $25^{th}$ to $75^{th}$ percentile of the model simulations, used to constrain the charcoal signal, were relatively constant with a small range, resulting in relatively constant fire emissions for this region (**Figure 11**).

The charcoal-based trend for EURO is based on records from both southern and northern Europe (Appendix B) and showed an increase from 1750 to present, whereas the model simulations in general showed no trend or a decrease from 1750 to present (**Figure 11**). Constraining the charcoal signal with the model output resulted in relatively constant fire emissions over Europe from 1750 to the present.

Central Asia (CEAS) is the temperate region, which contributed most to the global totals with 4.1%. Biomass burning emissions were based on the median of the models used. Most models, except ORCHIDEE, were relatively constant until 1950, after which emissions decreased. Using the median resulted in biomass burning emissions with a decreasing trend from 1750 to present (**Figure 11**).





**Figure 11: Fire carbon emissions for temperate regions. Panels on the left indicate all model outputs scaled to the average GFED values over 1997-2003 for that region. The panels on the right indicate the charcoal signal in green (solid line), the median of the models in purple (solid line) and the GFED signal in black. The variation between the models is shown in pink (25th to 75th percentiles) and light pink (total range models).**



## 3.3 Sensitivity analyses

Reconstructing fire emissions is difficult because there is very little data to constrain patterns and the existing data is often conflicting. In this section we describe the sensitivity of our results to some choices that had to be made rather arbitrarily, including choosing between which percentiles of the model outputs we scaled our results (4.3.1) and the choice of the fire models (4.3.2).

### 3.3.1 Effect of choice of percentiles

For the regions where we used charcoal as a proxy for fire emissions (Figure 4), we relied on the 25$^{th}$ to 75$^{th}$ percentile of the models to scale the charcoal signal (Section 2.4). If we had chosen the 5$^{th}$ to 95$^{th}$ percentile instead, global biomass burning emissions would have increased by 4.6%. This is mainly because TENA-E would have had more than six times higher fire emissions during the first part of our record because SIMFIRE results would be included (**Figure 11**). This would have increased the relative contribution of this region to the global total from 0.74% to 5.43% (Table 1). Europe (EURO) and eastern boreal North America (BONA-E) would decrease substantially, although those regions were relatively small contributors to the global totals. Western boreal North-America (BONA-W) and western temperate North-America (TENA-W) would also have decreased with a relatively small difference (-3.8% and -6.3% for BONA-W and TENA-W respectively) (Table 1).





**Table 1**: Average regional biomass burning emissions (1750-2015) and their relative contribution to the global total emissions. Numbers in parenthesis indicate estimates based on the 5[th] and 95[th] percentiles instead of the 25[th] and 75[th] percentile used throughout the study to scale the charcoal signal.

| | | Average emissions (Tg C year$^{-1}$) | Relative contribution (%) |
|---|---|---|---|
| BONA-W | Boreal North America – West | 41.1 (39.5) | 2.2 (2.0) |
| BONA-E | Boreal North America – East | 12.5 (10.7) | 0.7 (0.5) |
| TENA-W | Temperate North America - West | 8.4 (7.9) | 0.5 (0.4) |
| TENA-E | Temperate North America – East | 14.1 (107.7) | 0.7 (5.4) |
| CEAM | Central America | 44.5 | 2.4 |
| NHSA | Northern Hemisphere South America | 26.4 | 1.4 |
| ARCD | Arc of Deforestation | 53.6 | 2.8 |
| EURO | Europe | 7.0 (4.41) | 0.4 (0.22) |
| MIDE | Middle East | 3.1 | 0.2 |
| NHAF | Northern Hemisphere Africa | 475.4 | 25.17 |
| SHAF | Southern Hemisphere Africa | 623.3 | 32.9 |
| BOAS | Boreal Asia | 101.3 | 5.3 |
| CEAS | Central Asia | 78.2 | 4.1 |
| SEAS | South-East Asia | 207.3 | 10.9 |
| EQAS | Equatorial Asia | 47.3 | 2.7 |
| AUST | Australia | 97.4 | 5.1 |
| SARC | South of Arc of Deforestation | 51.3 | 2.7 |
| GLOBE | Sum of all regions | 1896.4 (1983.42) | 100.0 |



### 3.3.2 Impact of excluding models on regional emissions

We used six different models in our regional analyses, all with different temporal patterns. If new proxies become available, benchmarking exercises may indicate which models provide the most reasonable results but at this stage it is not known which models are best suited for our purpose. To better understand the sensitivity of our results to the selection of the models we tested what the effect would be on the average regional emissions over 1750-2015 if we excluded one of the six models (Table 2). The estimates from the ARCD and EQAS regions were not based on models and will thus not show any differences.

The effect on the average global totals by excluding models is relatively small (varying from -3% for excluding SIMFIRE to +1 or -1% for any other model). However, on a regional scale differences could be profound, with the largest differences again in temperate North America (TENA-E and TENA-W) where the models exhibited a relatively large range (**Figure 11**). In TENA-W excluding CLM would have increased the average emissions with around 35% and excluding INFERNO, JSBACH, LPJ-GUESS-SPITFIRE or ORCHIDEE would have increased the average emissions with 19-23%. In TENA-E, excluding INFERNO or JSBACH would have resulted in the biggest difference with increases of 42-44%, whereas excluding LPJ-GUESS-SPITFIRE or ORCHIDEE would have resulted in a decrease (both -35%). Another region where excluding individual models would have had a relatively large effect is eastern boreal North America (BONA-E), excluding SIMFIRE would have resulted in an increase in fire emissions of 21%. However excluding any other model would have resulted in a decrease, where excluding CLM, INFERNO and JSBACH had the largest effect (with a decrease around -20%). However on a global scale, TENA-E, TENA-W and BONA-W were relatively small contributors (Table 1).

In absolute terms emissions in SEAS, SHAF and BOAS were most influenced by excluding one of the models. In SHAF excluding SIMFIRE or ORCHIDEE would have had the largest effect resulting in a decrease of 20 Tg C yr$^{-1}$ excluding SIMFIRE or an increase of +20 Tg C yr$^{-1}$ excluding ORCHIDEE. Excluding one of the other models would also have had a substantial increase (CLM, JSBACH) or decrease (INFERNO, LPJ-GUESS-SPITFIRE), although the relative changes were relatively small (varying from -2% to +3%). In SEAS excluding one of the models would have resulted in either a



decrease varying from -12 to -14 Tg C yr$^{-1}$ (for ORCHIDEE, SIMFIRE and JSBACH) or a increase in the same magnitude varying from +12 to +14 Tg C yr$^{-1}$ (for LPJ-GUESS-SPITFIRE, CLM and INFERNO). Excluding one of the models in BOAS would have resulted in changes varying from +8 Tg C yr$^{-1}$ to 9 Tg C yr$^{-1}$ (JSBACH, LPJ-GUESS-SPITFIRE and ORCHIDEE) or -7 Tg C yr$^{-1}$ to -9 Tg C

5    yr$^{-1}$ (Table 2). In summary, our global numbers were rather insensitive to excluding one of the six models, but on a regional scale differences can be profound.

**Table 2**: Difference in average regional fire emissions (1750-2015) when a single model was excluded in absolute values (Tg C yr$^{-1}$) and as a percentage of the values used in this study. In ARCD and EQAS biomass burning emissions were not based on models.

| | SIMFIRE | | CLM | | Inferno | | JSBACH | | LPJ-GUESS-SPITFIRE | | ORCHIDEE | |
|---|---|---|---|---|---|---|---|---|---|---|---|---|
| | Average emissions (Tg C year$^{-1}$) | Relative contri-bution (%) | Average emissions (Tg C year$^{-1}$) | Relative contri-bution (%) | Average emissions (Tg C year$^{-1}$) | Relative contri-bution (%) | Average emissions (Tg C year$^{-1}$) | Relative contri-bution (%) | Average emissions (Tg C year$^{-1}$) | Relative contri-bution (%) | Average emissions (Tg C year$^{-1}$) | Relative contri-bution (%) |
| BONA-W | -1.0 | -2 | -2.3 | -6 | -3.3 | -8 | -4.2 | -10 | -3.1 | -8 | -3.8 | -9 |
| TENA-W | 1.1 | 13 | 3.0 | 35 | 1.6 | 19 | 1.7 | 20 | 1.9 | 23 | 1.8 | 22 |
| CEAM | -3.7 | -8 | 3.5 | 8 | 3.2 | 7 | -3.4 | -8 | -3.1 | -7 | 3.4 | 8 |
| NHSA | -3.3 | -13 | 2.9 | 11 | -3.2 | -12 | 2.7 | 10 | -2.4 | -9 | 3.2 | 12 |
| EURO | -1.2 | -18 | -0.3 | -4 | -0.6 | -9 | -0.5 | -7 | -0.2 | -2 | -0.4 | -6 |
| MIDE | -0.2 | -6 | -0.1 | -3 | -0.2 | -6 | 0.2 | 6 | 0.1 | 2 | 0.2 | 6 |
| NHAF | 0.2 | 0 | 5.1 | 1 | -6.5 | -1 | 3.8 | 1 | -6.8 | -1 | 4.1 | 1 |
| SHAF | -20.2 | -3 | 16.9 | 3 | -10.2 | -2 | 9.5 | 2 | -16.4 | -3 | 20.4 | 3 |
| BOAS | -9.4 | -9 | -7.4 | -7 | -9.1 | -9 | 9.3 | 9 | 9.2 | 9 | 7.5 | 7 |
| CEAS | -4.5 | -6 | 2.5 | 3 | -4.3 | -5 | 4.6 | 6 | -2.9 | -4 | 4.7 | 6 |
| SEAS | -14.5 | -7 | 14.2 | 7 | 12.7 | 6 | -12.3 | -6 | 14.4 | 7 | -14.5 | -7 |
| AUST | -4.5 | -5 | -4.3 | -4 | 3.9 | 4 | 1.9 | 2 | 2.8 | 3 | 0.2 | 0 |
| SARC | -2.0 | -4 | 0.8 | 2 | -0.3 | -1 | 1.2 | 2 | -1.6 | -3 | 1.9 | 4 |
| BONA-E | 2.6 | 21 | -2.5 | -20 | -2.7 | -22 | -2.6 | -21 | -1.8 | -14 | -1.9 | -15 |
| TENA-E | 1.4 | 10 | 3.4 | 24 | 6.3 | 44 | 6.0 | 42 | -5.0 | -35 | -4.9 | -35 |
| GLOBE | -59.3 | -3 | 35.4 | 2 | -12.7 | -1 | 17.8 | 1 | -14.9 | -1 | 22.1 | 1 |



## 4 Discussion

Carbon emissions increased slightly over the full time period and peaked during the 1990s after which they decreased gradually. Africa accounts for a large part (on average 58% over our study period) of global fire carbon emissions and the general trend therefore largely mimics that of Africa. The exception is the latter part of our record; from about 1950 African fire emissions decreased while emissions in deforestation zones increased (Figure 5). From 1960 onwards the interannual variability increased as a result of more detailed information from the visibility record for Equatorial Asia (EQAS) and the Arc of Deforestation (ARCD) and satellite-based biomass burning emission datasets covering the whole globe. This is thus partly an artefact of data availability but also partly real because the interannual variability from deforestation zones is relatively high and its contribution increased over time.

The multi-model median indicated that Southern Hemisphere Africa (SHAF) had an increasing trend from 1750 until ~1950, after which emissions stabilized, probably as a result of increasing $CO_2$ concentrations and changes in population density as input parameters. Regional studies based on charcoal show a decrease for African emissions from ~1900 onwards (Tierney et al., 2010). An explanation for this could be the intensification of agriculture, which suppresses fires in African savannas (Andela and van der Werf, 2014). Based on the relationship between cropland, burned area and precipitation found in Andela and van der Werf (2014) we reproduced fire emissions back to 1750, using cropland extent (1750-2014) from the Land Use Harmonization (LUHv2.2) dataset (Hurtt et al., 2011), in combination with MODIS MCD12C1 cropland for the year 2012. The reconstructed fire emissions based only on precipitation and changes in cropland as input variables showed similar results as the biomass burning emissions based on the median of models for both southern and northern hemisphere Africa from 1950 to 2013 (**Figure 12**). Although the trends for the two approaches over 1700-1950 agree for NHAF, in SHAF they show opposing trends with an increase from 1750-1950 based on models and a slight decrease based on the reconstruction.





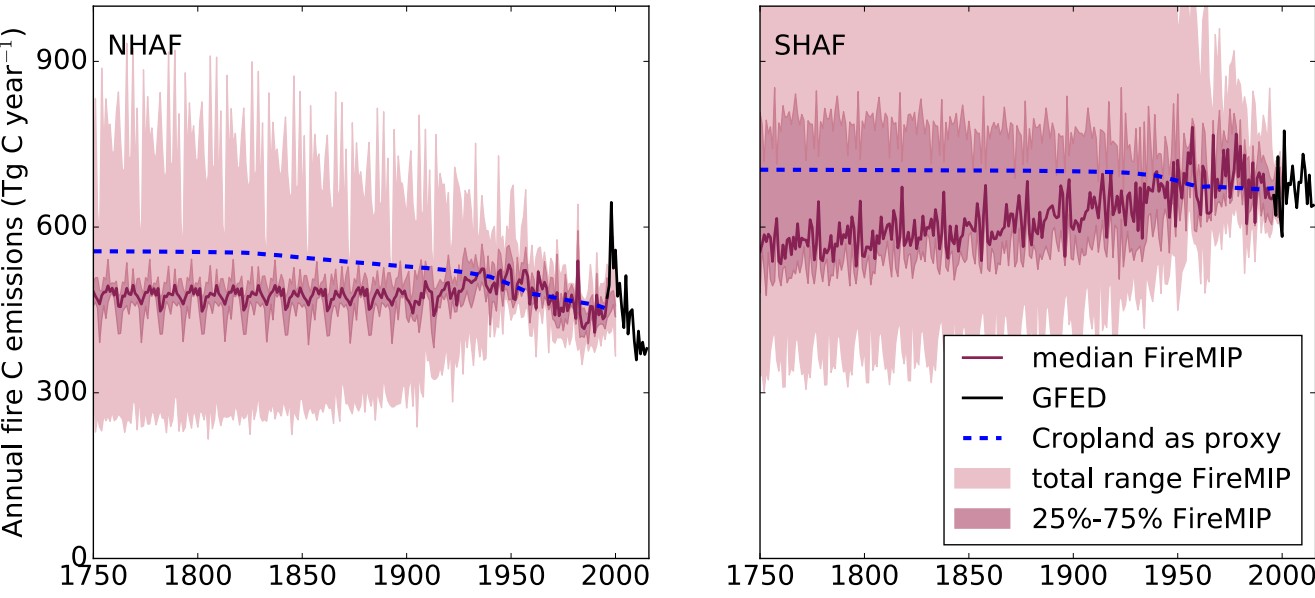

**Figure 12: African biomass burning emission estimates from 1750-2015 based on models, GFED and a model based on cropland change as proxy.**

Emissions from tropical forests are responsible for the global increase we found from 1950 onwards. Rainforests rarely burn in their natural state, due to their generally moist conditions underneath the canopy and because dry lightning is rare (Cochrane, 2003). Logging and land-use change made the landscape more vulnerable to fires (Nepstad et al., 1999). Infrastructure projects, including the building of roads and highways, increased the migration into the Amazon basin (Fearnside, 2002; Laurance et

al., 2001), but also, for example, the Mega Rice Project during the 1990s where peatland drainage in Kalimantan increased fire emissions in EQAS (Field et al., 2009). Before humans substantially altered the landscape, we assumed that fire emissions did happen, either man-made or naturally, but at a much lower rate. Interannual variability in tropical regions is partly driven by changes in the El Niño Southern Oscillation (ENSO) for both South America and Indonesia, and the Indian Ocean Dipole (IOD) for

Indonesia (Chen et al., 2013; Field et al., 2009).

Over the past decade, several studies have identified larger variability or trends over our study period than we present here. This includes a steeper increase of global fire emissions from 1750 to 1920 than found by us, after which fire emissions gradually decline from 1920 to present based on a global





analysis of the charcoal record (Marlon et al., 2008) and much larger variability based on CO concentrations and their isotopes from a South Pole ice core (Wang et al., 2010). Our results are different than the patterns found when relying solely on charcoal data, because we limited ourselves to that approach for regions where the density of charcoal was relatively large. However, all regions have

charcoal records (Appendix B) and results would have been somewhat different had we used those. The variability we found is smaller than found in the CO record (Wang et al., 2010) and despite that their pattern is difficult to reconcile with our current understanding of fire emissions and atmospheric transport (van der Werf et al., 2013). Other sources of information include the use of $CH_4$ concentrations in ice cores (Ferretti et al., 2005) and firn air samples, although it is uncertain to what

degree the most recent part of the record is representative for current conditions. These studies show an increase over the recent decades for both the Northern (Wang et al., 2012) and Southern Hemisphere (Assonov et al., 2007), and at this point we cannot reconcile the differences found in the various records indicating that uncertainty remains substantial.

### Comparison with CMIP5 estimates

The biomass burning emissions used in the Coupled Model Intercomparison Project phase five (CMIP5) and available for 1850 through 2000 were estimated using GFED version 2 for 1997 onwards and biomass burning inventories (GICC and RETRO) for the pre-satellite era. Biomass burning emissions were kept constant from 1850 to 1900 based on the 1900 value, which was lower than current emissions. From 1900 to 1920 emissions decreased, after which they increased rapidly to 2000 (Figure

13, Lamarque et al., 2010). Our results show a somewhat smaller amplitude for most species and less of an increase, although differences vary depending on the specie one is interested in due to the use of revised emission factors and the relative contribution of forest fires (emitting in general high amounts of reduced gases such as CO and low amounts of $NO_x$) versus savanna fires (low CO, high $NO_x$) (Figure 13).





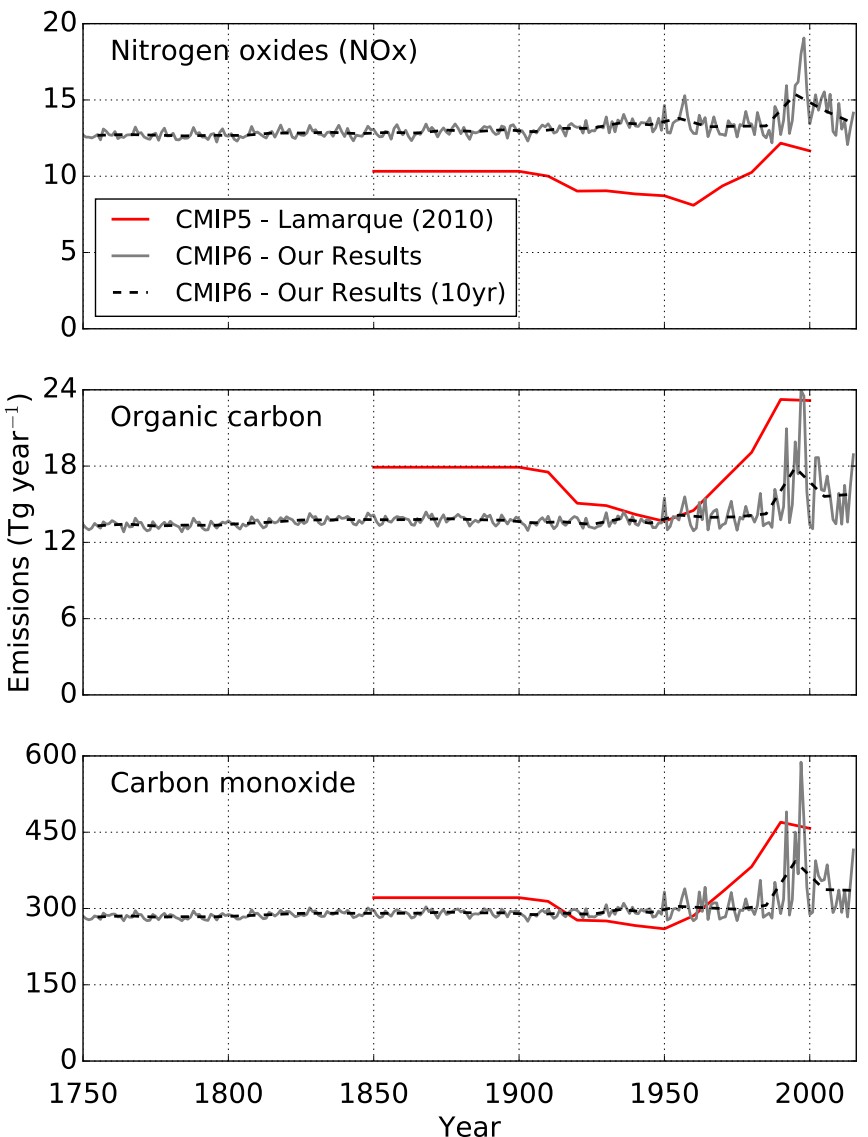

**Figure 13:** Total global biomass burning emissions for NOₓ, organic carbon, and carbon monoxide estimated by Lamarque et al. (2010) developed for CMIP5 and our results developed for CMIP6 on an annual and decadal time step.

## 5 Uncertainties

Uncertainties in reconstructing fire emissions are large and stem from uncertainties in the data we used and from our approach of combining the different datasets. For the reconstruction the fire models and visibility-based fire emissions were used with GFED4s as an anchor point. We have relied on fire



models in almost every region, except ARCD and EQAS. The fire models exhibit differences in regional trends, resulting in a range in regional biomass burning emissions. On a global scale, the impact of excluding single models led to relatively small differences up to 3% (Table 2). However, on a regional scale differences were more profound, with percentages up to 44% in TENA-E. In regions where models were used in combination with charcoal records, the models had a large influence when the charcoal signal and the models exhibit opposing trends, for example in EURO and BONA-W and this also explains why in these regions excluding any of the models would result in a decrease in fire emissions (Table 2).

Given the good agreement between visibility and GFED estimates for the overlapping period in ARCD and EQAS we feel these regions are relatively well represented. However, this proxy relies on observations of humans with inconsistencies and also the location of the WMO stations were not necessarily evenly distributed over the region. Also, little is known about fire history in these regions before visibility-observations became available. We have assumed that fire emissions did happen, either man-made or naturally, but at a much lower rate but in many regions complicated relationships were found (Archibald, 2016).

Over the 1997-2015 period we used fire-emissions based on GFED4s. In that approach burned area, fuel consumption, and emission factors all have uncertainties although each parameter has seen important improvements over the past decade. The inclusion of small fires has increased burned area in human-dominated locations and total burned area now better agrees with higher resolution burned area in several regions (Mangeon et al., 2015; Randerson et al., 2012). The fire distribution in regions with small fires from, for example agricultural waste burning, now also agrees better with those in inventories derived from active fire observations (Chuvieco et al., 2016). However, more systematic comparisons are necessary to assess the exact uncertainty in this approach. Likewise, modelled fuel consumption has benefited from comparisons against field measurements compiled by van Leeuwen et al. (2014) and modelled and measured values are now in good agreement on biome level, but comparisons within biomes still show substantial differences (Andela et al., 2016; Veraverbeke et al., 2015). Finally, the emission factors used here from Akagi et al. (2011) distinguishes more classes (for example boreal and temperate regions which were previously lumped together) and the various studies



are dealt with in a more systematic way than previously, but for many species measurements are lacking and to date we still do not understand well the spatial and temporal variability of emission factors, especially within biomes (Knorr et al., 2012; van Leeuwen and van der Werf, 2011).

GFED fire emissions were also used to distribute the regional annual fire emissions in space and time in the pre-GFED time period based on the 1997-2015 climatology. This approach ignores variability due to changes in fire weather and land use. For example in Africa, where many savanna regions have been converted to agricultural land (Andela and van der Werf, 2014) and in EQAS and ARCD where dense tropical rainforest is converted to small-scale agriculture and large-scale industrial agroforestry including infrastructure (Cochrane and Laurance, 2008; Field et al., 2009; Laurance et al., 2001), the spatial pattern has changed over time which is not accounted for in our approach.

In this study we have used a regional approach by merging several data sources. There is still much to be gained by collecting more data and using different species. Levoglucosan, for example, is a biomarker for fires and Kehrwald et al. (2012b) showed that levoglucosan in a Greenland ice core represents the fire signal from Asian and North-American source regions. Other proxy records that could improve regional estimates are char and soot measurements taken from löss. These can be used to validate the estimates in CEAS (Han et al., 2010). As the Global Charcoal Database continues to evolve with new data contributions, regions that are currently under sampled could inform GCD-based biomass burning histories. Finally the FireMIP exercise may lead to a better representation of the processes driving global fire patterns, which itself will help in developing a more complete understanding of fire since the year 1750. For a rough indication of uncertainty in these regions, Figure 14 shows comparisons between our results, charcoal Z-scores (1750-2000) from GCDv3, and burned area reconstruction by Mouillot and Field (1900-2000) for Sub-Saharan Africa, Patagonia, Boreal Asia and South East Asia (Appendix B). The three datasets quantify fire histories using different units, so all datasets were scaled and transposed to the year 2000 value to qualitatively compare the trends. In Sub-Saharan Africa, CMIP6 and GCv3 are similar from 1950 to present, but CMIP6 decreases more rapidly prior to 1950 (Figure 14). The trend in Boreal Asia also agrees for a large part, where charcoal estimates exhibit a larger range in variation over time. In Patagonia and South East Asia, the general trend is increasing, although the peak years differ. To improve and constrain our dataset, we encourage paleo-





fire researcher to sample their sites in detail for the last 250 years, even though proxy-records are currently mostly used for longer (century to millennial) time scales. Pinpointing the reasons behind outliers and opposing trends between the various models will lead to lower uncertainties for studies like ours.

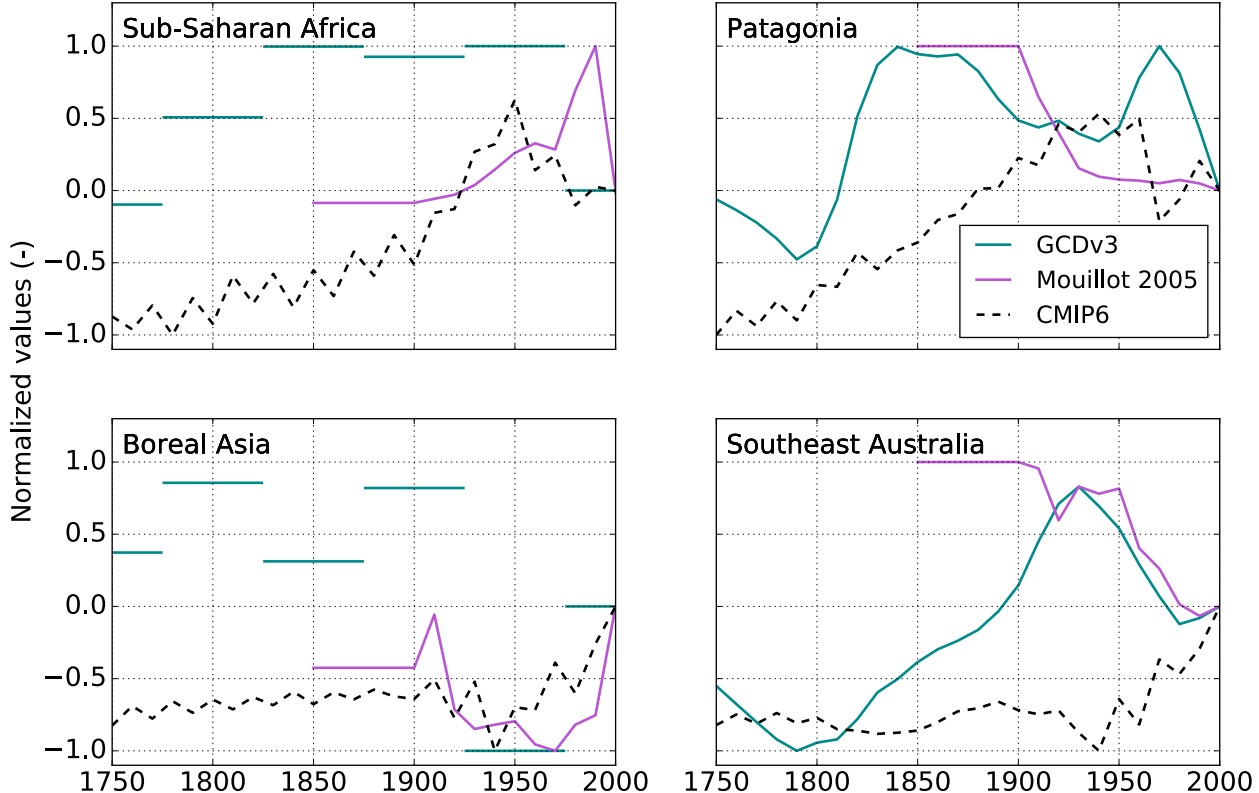

**Figure 14:** **Normalized Z-scores of Charcoal (1750-2013, blue), normalized decadal emissions based on our estimates (1750-2000, black) and normalized emission estimates based on Mouillot and Field (2005) (1850-2000) for Sub-Saharan Africa, Patagonia, Boreal Asia, and Southeast Australia (regions outlined in Appendix B). For Sub-Saharan Africa and Boreal Asia charcoal is based on 50 year windows, whereas for Patagonia and Southeast Australia 10 year windows were used.**





**Guidance for using this dataset as forcing in climate models**

This dataset is made available as forcing dataset for the Coupled Model Intercomparison Project Phase 6 (CMIP6) analyses at the PCDMI repository (https://pcmdi.llnl.gov/search/input4mips). The emissions for BC, $CH_4$, CO, $H_2$, $N_2O$, $NH_3$, NMVOC, $NO_x$, OC, and $SO_2$ were provided. The NMVOC emissions

consists of the sum of $C_2H_6$, $CH_3OH$, $C_2H_5OH$, $C_3H_8$, $C_2H_2$, $C_2H_4$, $C_3H_6$, $C_5H_8$, $C_{10}H_{16}$, $C_7H_8$, $C_6H_6$, $C_8H_{10}$, Toluene_lump, Higher_Alkenes, Higher_Alkanes, $CH_2O$, $C_2H_4O$, $C_3H_6O$, $C_2H_6S$, HCN, HCOOH, $CH_3COOH$, MEK, $CH_3COCHO$, $HOCH_2CHO$. These NMVOCs are also provided separately. These are total emissions, ancillary datasets with contribution of emissions related to agricultural waste burning, fires used in deforestation, boreal forest fires, peat fires, savanna fires and temperate forest

fires are provided.

Climate models should not use the $CO_2$ emissions (or nitrogen emissions if the nitrogen cycle is included in the model) as forcing because in general these emissions are not net emissions to the atmosphere, but a return pathway of previously sequestered carbon, just as respiration is. The exceptions are $CO_2$ emissions from deforestation and peat fires. However, the models that do not

simulate land use change are recommended to use land use change emissions prepared for AR6 (http://www.mpimet.mpg.de/en/science/the-land-in-the-earth-system/working-groups/climate-biogeosphere-interaction/landuse-change-emission-data/). Models that have their own fire model but do not simulate anthropogenic fires are advised to use only the emissions related to deforestation and agricultural waste burning. We provide the fraction of emissions associated with this. While the large

interannual variability is a key feature of global fire emissions, modelers may consider averaging out this fire signal to avoid having interannual variability in fires being out of sync with interannual variability in climate.

**5 Conclusion**

We have merged satellite-based fire emissions for recent times, charcoal datasets in temperate and

boreal regions, visibility-records from weather stations over tropical forest regions, and emission estimates from the FireMIP project. Our aim was to make the best use of the strengths of the various



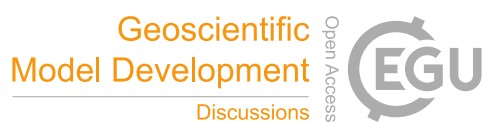

datasets using a regional approach. According to our estimates, global biomass burning carbon emissions increased slightly over the full time period and peaked during the 1990s after which they decreased gradually. The global pattern varies somewhat depending on trace gas or aerosol species. Africa accounts for a large part (58%) of global fire carbon emissions and the general trend therefore

mimics that of Africa especially in the early part of our record. African fire emissions exhibited a decrease from 1950 onwards as a result of conversion of fire-prone savannas to agricultural land. The absence of pre-industrial fire history data in Africa in particular, is a major limitation of these estimates. This decrease in Africa is partly offset by increasing emissions in deforestation zones especially during the 1990s which also led to higher interannual variability in fire emissions. Our results point towards

less variability over time than the fire emissions used in CMIP5 and a smaller difference between pre-industrial and present emissions.

**6 Code availability**

The Python code that was used to assimilate the raw data is available on author request.

**7 Data availability**

This dataset is made available as forcing dataset for the Coupled Model Intercomparison Project Phase 6 (CMIP6) analyses at the PCDMI repository: https://pcmdi.llnl.gov/search/input4mips. GFED4s data is publicly available at http://www.globalfiredata.org/data.html. Charcoal records are available through the Global Charcoal Database (https://www.paleofire.org). Regional visibility-based fire emissions and regional emissions based on the different fire models can be requested via the corresponding author.



## 8 Appendices

### 8.1 Appendix A: Description and application of the SIMFIRE-GDP model

In its coupled version, LPJ-GUESS-SIMFIRE uses SIMFIRE to compute burned area based on a stand-alone semi-empirical model optimised against current observations (Knorr et al., 2014), and LPJ-GUESS to computer vegetation dynamics, the biogeochemical cycle (Smith et al., 2001), fire impacts according to Knorr et al. (2012), and a coupling scheme between SIMFIRE and LPJ-GUESS described by Knorr et al. (2016) Different to the original version of SIMFIRE, the present version uses regional averages of per capita gross domestic product (GDP) in addition to human population density as statistical drivers to compute burned area, in addition to climate and vegetation factors. The following non-linear predictor was inverted against GFED3 observed burned area in the same way as described by Knorr et al. (2014), on a global 0.5 by 0.5 degree grid excluding croplands:

$$A(y)=a(B)F^b N_{max}(y)^c \text{logit}(d+ep+fGp) \qquad (A1)$$

$A$ is fractional burned area (in $yr^{-1}$), $B$ is biome type, $F$ is the multi-year average of the annual maximum fraction of plant-available photosynhetically active radiation (FPAR) derived from satellite observations (Gobron et al., 2010), $N_{max}$ the annual maximum Nesterov index divided by $10^5$ computed with observed climate data (Weedon et al., 2011), $p$ population density in $km^{-2}$ based on HYDE 3.1 for 2005 (Klein Goldewijk et al., 2010) , $G$ growth-domestic product per capita in 1995 US$ divided by $10^4$ where per capita GDP data were taken from HYDE 3.1 for 1995 and the per capita GDP of a grid cell equals that of the region to which the grid cell belongs, and $y$ fire year (which starts in a different month at each grid cell before the start of the fire season in the respective grid cell). logit is the logistic function with $\text{logit}(x)=1/[1-\exp(-x)]$. GDP data were available for the following regions: Canada, USA, Central America, South America, North Africa, Eastern Africa, Southern Africa, West Africa, OECD Europe, Former Soviet Union, Eastern Europe, Middle East, South Asia, Oceania, Japans, and Southeast Asia. Model inversion was carried out for all grid cells simultaneously optimizing a set of 13 free parameters against annual gridded fractional burned area. The optimal values were $2.32\times10^6$, $1.12\times10^6$, $0.76\times10^6$, $1.40\times10^6$, $6.27\times10^6$, $10.0\times10^6$, $0.38\times10^6$, $1.69\times10^6$ for $a(1)$ to $a(8)$ for the eight biomes, and for the global parameters $b=1.007$, $c=0.75$, $d=-16.0$, $e=0.0021$, and $f=-0.46$.



Using the coupled LPJ-GUESS-SIMFIRE (Smith et al., 2001) global dynamic vegetation–wildfire model, (Knorr et al., 2016) have found that at least for the first half of the $20^{th}$ century, climate factors had only a small influence on wildfire emissions, but that the main driver was population density. For extrapolating burned area back in time before 1901 only the part of Eq. A1 that relates to human factors

was used. The optimization of the SIMFIRE-GDP model thus yields a scalar function describing the impact of population density and GDP on fractional burned area, which is

$$P(G,p)=\text{logit}[-16+(0.0021-0.46G)p]/\text{logit}(-16). \qquad (A2)$$

This scalar $P$ has been normalized to yield a value of one in the absence of human activities and therefore describes the degree of human fire suppression. $P$ describes increasing burned area with

population density for low GDP, and vice versa for high GDP. GDP data is used for every five years from 1890 to 1995. Before and after that date, we keep per capita GDP per region constant in time. HYDE 3.1 population density values from 1700-2000 at a decadal scale were used. Furthermore historical HYDE 3.1 cropland fraction from 1700-2003 (Klein Goldewijk et al., 2011) was used to correct SIMFIRE estimates, setting wildfire emissions for croplands to zero.

To obtain fire emissions spanning the period 1700 to 2000, LPJ-GUESS-SIMFIRE was run using daily observed climate data (Weedon et al., 2011), yielding annual emissions for the period 1901 to 2000. Emissions for 1700 through 1900 are constructed by multiplying climatological emissions from the early $20^{th}$ century with a scalar $s$ defined as $s=P*f_c$, where $f_c$ is the cropland fraction. This scalar described the degree of human suppression of burned area as a function of population density, GDP,

and cropland fraction. Using $E_1$ as the average annual emission rate computed from te LPJ-GUESS-SIMFIRE during 1901 to 1930. $E_0$ as the 1901-1930 annual average emissions computed from a separate LPJ-GUESS-SIMFIRE simulation with population density set to zero (no-population emissions), $s_1$ as the temporal average of $s$ during 1901-1930, $f=(s-s_1)/(1-s_1)$, $x$ for location, and $t$ for time in years, we compute emissions prior to 1901 as follows:

$$E(t,x)=\begin{cases} E_0(x)\,s(t,x)\,/\,s_1(x) & \text{if } s<s_1, \\ f(t,x)\,E_0(x) + [1-f(t,x)]\,E_1(x) & \text{else.} \end{cases} \qquad (A3)$$




## 8.2 Appendix B

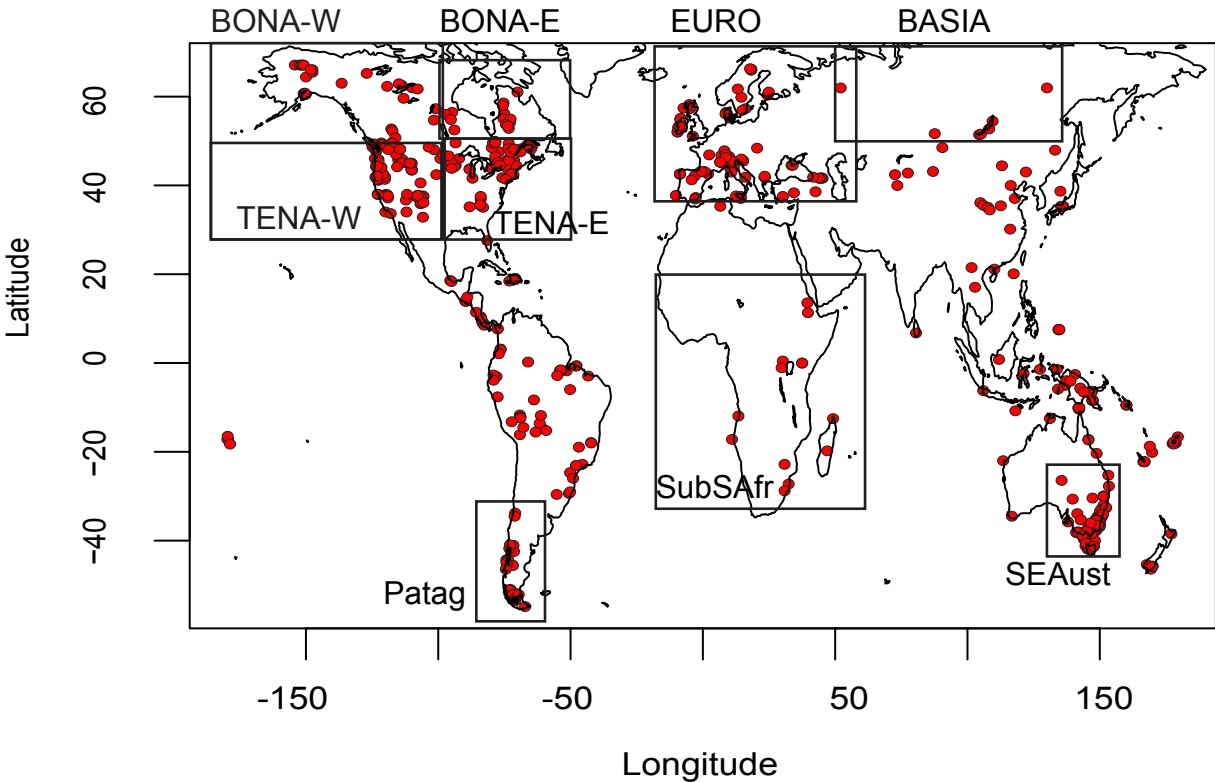

**Figure B1: Map with charcoal site locations (red dots) that have samples over the last 250 years and regions (black squares) used in this study.**



## 9 Acknowledgements

We would like to thank Kees Klein Goldewijk, Benjamin Aouizerats, Niels Andela, Pierre Friedlingstein and Thijs van Leeuwen for their help and useful discussions. This work was invited by Claire Granier, Steven Smith, the CMIP community, and the Interdisciplinary Biomass Burning

5    Initiative (IBBI). Furthermore we acknowledge the PAGES Global Paleofire Working group for making the Global Charcoal Database publicly available and supporting fire workshops. MvM and GvdW were supported by the ERC (grant number 280061). NSF award BCS-1437074 and BCS-1436496 provided workshop support and funding for JM and BM. PICS CNRS 06484 provided workshop support for ALD. RF was supported by the NASA Atmospheric Chemistry Modeling and Analysis Program. SH

10   and AA acknowledge support by the EU FP7 projects BACCHUS (grant number 603445) and LUC4C (grant number 603542). JK was supported by the EU H2020 project MACC-III (grant number 633080).





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
