# Peer review of "Historic global biomass burning emissions for CMIP6 (BB4CMIP) based on merging satellite observations with proxies and fire models (1750-2015)"

_Geoscientific Model Development, 2017_

## Short Comment (SC1) · 24 Feb 2017

Dear authors,

In agreement with the CMIP6 panel members, the Executive editors of GMD would like to establish a common naming convention for the titles of the CMIP6 experiment description papers.

The title of CMIP6 papers should include both the acronym of the MIP, and CMIP6, so that it is clear this is a CMIP6-Endorsed MIP.

Good formats for the title include:
'XYZMIP contribution to CMIP6: Name of project'

or

'Name of Project (XYZMIP) contribution to CMIP6'

If you want to include a more descriptive title, the format could be along the lines of,

'XYZMIP contribution to CMIP6: Name of project - descriptive title'

or

'Name of Project (XYZMIP) contribution to CMIP6: descriptive title.'

When you revise your manuscript, please correct the title of your manuscript accordingly.

Additionally, we strongly recommend to add a version number to the MIP description. The reason for the version numbers is so that the MIP protocol can be updated later, normally in a second short paper outlining the changes. See, for example: http://www.geosci-model-dev.net/special_issue11.html,

Yours,

Astrid Kerkweg

---

## Referee Comment (RC1) · Anonymous Referee #1 · 9 Mar 2017

This paper provides a description of the biomass burning emissions that are provided for the upcoming CMIP6 simulations. The authors have done an excellent job of providing in-depth description of the methodologies used to generate the emissions. This was a gargantuan task and the authors should be congratulated to achieving this. I have a small number of minor comments below. My main complaint is that the emissions are showing fairly significantly different trends from the CMIP5 dataset and it would have been very useful if some model simulations (or at least estimates of radiative forcing) had been performed to understand the consequences of these different trends. I understand that this probably beyond the scope of this paper, but it is still a shortcoming worth mentioning.

Minor comments

Page 2, line 23: CMIP is not part of IPCC. It is part of WCRP (see Eyring et al., GMD, 2016)

Page 11, line 13: how large was the scaling when applied? Might be good to mention the scaling algorithm (Eq. 1) at this point. Since 1997 was such a large emission year, has its role been evaluated?

Section 2.3: it seems that it would be useful to have more details on the methods used to extract emissions from visibility data? How does this work in anthropogenically-polluted areas?

Page 17, lines 16-27: any suggestions on how models could integrate that recommendation?

Page 18: change link to emission factors to an actual description in supplement. Web link will break over time

Comparison with CMIP5: it would be greatly helpful if regional comparisons were also shown, maybe simply in the supplemental material

———————————————————

---

## Referee Comment (RC2) · Anonymous Referee #2 · 13 Mar 2017

This paper presents the methods and results from the development of a global fire emissions inventory representing 1750-2015. The results are to be used as consistent inputs to climate model simulations. The authors integrate the results of fire models, satellite-based fire inventories, fire proxies (i.e., charcoal records), and visibility observations to provide emission estimates. The description of the methods is very complete. Further, this type of effort is incredibly challenging, and the authors provide an good discussion about the uncertainties in the assumptions they made in their approach. Despite shortcomings in the data and models, this is a very good effort and will provide improvements to future model simulations. I only have minor suggestions and some editorial comments for the authors.

[Figure]

General Comments: I may have missed this, but I would assume that the Fire models that are described need to be forced with atmospheric inputs. It is unclear to me what forcing were applied in the simulations that produced the emission results. This should be made more clear somewhere in the paper. I am assuming that they were all driven by the same climatic drivers?

El Nino is obviously an important driver of fire activity and emissions, particularly in EQAS. This is not captured in the emission estimates before the 1970's. Is this a problem? Can the authors comment on this further?

Editorial Comments:

Page 3, line 16: Should it be "directly" and "indirectly"

Page 3, line 24: What other land surfaces? The previous sentence talks about defor-estation fires. So, is this land surfaces other than forests?

Page 5, line 7: Change to "All of these"

Page 5, line 14: Change "which" to "that"

Page 6, lines 7-11: This is a very long sentence and could be broken up to read more clearly.

Page 6, line 18: The differences "over the past decade"? What is meant by this?

Page 7, lines 9 and 10: "data" are plural. Change to "provide"

Figure 1: Shouldn't the satellite observations circle expand downward to local scales?

Page 9, line 6: The reference for GFED4s should be provided.

Page 10, line 11: A comma should be used before the word "which" (here and through-out the paper).

Page 11: The emission factors used in this are from Akagi et al. 2011. Did you include the emission factors from the updates to this dataset (from 2015)

Page 14, line 17: Change "which" to "that"

Page 14, line 25: change the tense to be consistent ("are" should be "were")

Page 16, line 3: How can you compare the visibility outputs to 1750 – 2000 when those data don't go back that far? This is unclear.

Page 17, line 2: all "of" our

Page 18, line 22: Define IAV when first used.

Page 18, line 24: Change "which" to "that"

Page 28, line 2: Change to "there are very little data"

Page 34, line 18-19: Current emissions? Does this mean the current emissons (2000? 2010?) in the CMIP5 estimates?

Page 34, line 21: Should "in" be "is"?

Page 36, line 14: This sentence is worded poorly and should be rewritten.

Page 39, line 8: Should there be an ; or : after "emissions"
* * *

---

## Referee Comment (RC3) · Anonymous Referee #3 · 16 Mar 2017

The paper aims at providing a historical reconstruction of fire emissions from 1750 onward, as the basis for the CMIP6 climate modeling objective. This paper then focusses on updating the 1850-2000 fire emissions proposed for the CMIP5 exercise. To reach this goal, the authors use the GFED4s emissions data as the baseline for 1997 to present period. The backward trend line for the Tropical forest is based on newly delivered papers reconstructing fire emissions since the 1960's based on visibility indices. The global charcoal database is used for boreal and temperate forest of the northern hemisphere where the network of sample is the most significant and from a panel of DGVMs runs for the 1750-present period for all the other areas.

The objectives are timely, and the effort in assembling the state-of-the-art modelling

and charcoal communities deserves congratulations for proposing a synthesis. The strength of the paper in assembling 6 models, and readjusting the non-quantitative charcoal temporal variations to fit the final GFED4s time series, might also be however its main weakness. It is on one side a huge data assemblage, and on the other side a poorly investigated model intercomparison weakening the final message. Despite being well and clearly described, some assumptions remain confusing and potentially misleading. The total absence of link and usage of the MIP5 reconstruction is also frustrating.

The main assumption of the paper is that "fire models can be used to estimate biomass burning emissions on a global scale"(P4l21-23) , and this also on a long temporal scale. In this sense, the paper contradicts itself when, in the end, comparing model's performances to charcoal data on selected regions, and concluding on poor relationships. In absence of any other data, we might understand however to rely on this data ressource. I have listed below the questions I am concerned with, which would require major corrections and significant additional information. Unfortunately, I think this approach would really deserve a deeper FIREMIP result understanding before being used for this purpose.

When going through the 3 main methodological tasks used for the reconstruction, I have the following questions:

1. Visibility: this interpolation based on two published papers linking visibility to GFED emissions for the period 1997-present and extending backward to 1960's in south east asia and Amazonia is really convincing, both in terms of temporal trend and interannual annual variability. In this sense, this is a significant update to the MIP5 reconstruction. It would be interesting though to have this comparison with MIP5 for all regions, to clearly understand the added value of this synthesis (as performed in figure 14). I have just a little concern that the Van Marle et al. (2017) paper used for this reconstruction analysed only a portion of the ARCD region showed in figure 2. Peru and Eastern Brazilian (fire-prone cerrado savannas) don't seem to be included in this temporal trend

reconstructed from visibility. =>How did the authors deal with this other part of the ARCD region, still representing a significant surface?

2. charcoal-based reconstruction The authors used the global charcoal database, providing a general trend in historical charcoal deposition in sediments from vegetation fires, with increasing time resolution allowing for decadal understanding of fire history. The authors selected the regions with a significant amount of data, which is a fair assumption. The main weakness of this dataset is the missing quantitative information so the authors had to rescale the Z-scores of the charcoal database to the emissions. The method is described in p17. We get a little confused p17l9-10 with the sentence "the normalized charcoal signal (CCnorm) is the unitless charcoal influx Z-score on a decadal time step normalized per region and year Âż. this is minor, but decadal and yearly time step sound confusing to the reader. That should be rephrased. When looking at Power et al. 2010 and Marlon et al. 2016 papers, Z scores vary below 0 and above 1, so I guess these values have been reduced to the 0-1 interval. Is that correct? Maybe rephrase as we understand, as written, that Zscores are directly between 0 and 1 in the raw data. To rescale the Z score, the authors then assume that the maximum Z-score corresponds to the 75th percentile of FIREMIP models and the minimum z score to the 25th percentile in equation 2. This assumption is then thoroughly and properly discussed later. We wonder however in Equation 3 p17, why CCscaled is based on CCfireMIP of the year 2000 and not the mean 1997-2003 period as FIREMIPscaled (equation 1)? The output from this rescaling is finally a 10-year smooth average, without any interannual variability (as shown in figure 10 for example). Then why not using the FIREMIP interannual variability to produce this missing variability on the smooth charcoal trend? For the EU region, the charcoal database is used. Samples are distributed across Europe, while burned area is mostly located in the south on the mediterranean part. Are the charcoal sample locations weighted according to present observed burned area for exemple to give more weight to the Mediterranean? If not how biased could be the result? For north America, The method is clearly described and discussed so that could be convincing. I still wonder here,

however, why the authors did not use the forest fire statistics from US and Canada and reconstructions of burned areas going back in time for almost a century in these regions widely documented to rescale the minimum and maximum emissions? These data have been used in MIP5 and in my opinion would have greatly benefited here to strenghthen the decision of this 25th and 75th percentile, and make a link to the previous version.

3 DGVMs historical runs In absence of any substantially reliable information, the authors decided to use the FIREMIP runs. The choice is clearly stated in the methods. It then covers a very significant portion of the globe (Africa, south America beside Amazonia, Asia, and Australia) and a large portion of the global burned area. Figure 3 could be rearranged proportionally to burned area, so that the reader clearly visualize that the global burned area reconstruction relies mostly (round 75% ) on models... I am not against this idea, but in turn, the reader is left a little disappointed and questioned as the paper doesn't analyse at all models assumptions and specificities. The authors give us the huge variability from the models (which is disappointing but actually in the range of uncertainties in climate model projections) and we don't really know what is climate-driven, human-driven and why each model has this trajectory. Analyzing all this would require one full (or even several) papers from this modelling group so they give us further information. and it's a huge task. I might understand the rush to provide CMIP6 data for burned area emissions, but this chapter leaves the reader very frustrated, if not suspicious on the reliability of these data for this purpose. I guess the authors would argue that it's still better than the empirical reconstruction from MIP5 and the linear trend used before 1900.when looking at figure 5 and the 1750-1900 trend, it's not obvious that the authors have achieved a fundamentally innovative trend compared to MIP5.

When going into details on this chapter, I have the following questions:

- P12 l2: FIRE MIP runs DGVMs from 1700 to 2013. GFED from 1997 to 2015. The overlapping period is 1997-2013. Why using 1997-2003 further on (line 5) as an overlapping period? - timing of interannual variability: I was expecting that, if the trend is not overwhelmingly different from the flat trend of MIP5, we would get the actual interannual variability in time and amplitude from this approach. We also get a little disappointed as all experiments used repeated 1901-1920 forcings from the beginning of the simulation (1750) to 1900. In this sense, figure 5 is misleading and should better be presented as a moving window decadal values with uncertainties (SE or coeff of variation), as the variability is not timely. Also why minimizing interannual variability ( P12 L12-L14) on purpose? The authors in additions discuss about the increasing interannual variability but the trend of this variability in figure 5 is all fake. This should not be taken for granted as: a) considering the mean when emission simulations are not timely in phase for each model (figure 7 for example) intrinsically reduces the interannual variability (lower than each model's interannual variability) , b) the charcoal time serie is flat (discussed above). Why do the authors provide this 'fake' interannual variability ? is that a request from the CMIP6? It would be worth, in the introduction for exemple, to present the CMIP6 'wish list' to better understand the choices perfomed in this reconstruction. We are also questioned that the authors used the 25th and 75th percentiles for charcoal reconstruction using FIREMIP models , so that "outliers did not influence the scaled regional charcoal signal" (P15L15). We then wonder why this was not also done for equation 1. In conclusion for this modelling chapter, if we can knowledge the effort of the authors to assemble all this information, the conclusions seem way too overrated and we miss a lot of the understanding of this model intercomparison to fully appreciate the synthesis. The interannual variability is an important point that is completely misrepresented in the final results and misleading for the readers.

Discussion: The discussion is interesting and actually provides more interesting information than the results themselves. However, it also highlights the weakness of the results. P32 l1: we wonder if the visual trend is actual or driven by the "fake" interannual variability. Statistical time series analysis could reinforce this sentence, but with a wrong interannual variability they will be also biased.

P32 l13-14: "after which emissions stabilized, probably as a result of increasing CO2 concentrations and changes in population density as input parameters" This sentence clearly illustrates my comments on the poor analysis of the models functioning. It is very difficult here to understand and have an opinion based on the information provided in the paper (neither by reading hantson et al 2016 and Rabin et al describing the models): why increasing CO2 would stabilize fire emission? For SAH, different trends are observed in models. . .but all are driven by population (at least ORCHIDEE and LPJ GUESS SPITFIRE are coupled with the same SPTIFIRE but with the most opposite trends. . .). A full model output analysis would be worth being published before this paper, to strengthen the message.

Figure 13 p 33: Using the Andela and van der Werf (2014) hypothesis seems to be a fair option to reconstruct fire history actually for Africa. That's a nice result. Why not choosing this trend the same way the authors did with charcoal? This would completely reverse the global increasing trend obtained from the FIREMIP into a decreasing trend, and would fit the charcoal Tierney (2010) trend. That sounds convincing. How is cropland area introduced in DGVMs? If not included, there is no reason to value the model hypothesis rather than the Andela paper. This paragraph is again both exciting as the authors seem to have found a smart proxy fitting the charcoal but they don't use it, but also disappointing as it weakens the model's approach, that we are not able to fully appreciate due to a lack of deep analysis.

The final discussion chapter on the comparison with MIP5 is welcome (at last!). Too bad it's partial and only focused on few areas. A final comparison on the MIP5 and MIP6 would be also interesting. . . as the MIP6 seems to be flat before 1900, and it sounds like it would be very similar to MIP5 in the end.

Some few minor additional comments: P3L8 : the varying constraint hypothesis from krawchuk and moritz 2011 would be a better reference in addition or replacement of van der werf 2008. P4l21-23: this is a critical assumption that "fire models can be used to estimate biomass burning emissions on a global scale" on a historical point of

view... maybe review some recent papers trying to compare historical trends (Yue et al., Kloster et al., Yan et al.). P18 l 22: IAV? Does it mean interannual variability?

P38: figure 14: just wondering if charcoal Z-scores should be rescaled to the 50 year average of burned area from Mouillot & field and C emissions from your study to better rescale the temporal trend, instead of year 2000.

---

## Author Comment (AC3) · 15 May 2017

This paper presents the methods and results from the development of a global fire emissions inventory representing 1750-2015. The results are to be used as consistent inputs to climate model simulations. The authors integrate the results of fire models, satellite-based fire inventories, fire proxies (i.e., charcoal records), and visibility observations to provide emission estimates. The description of the methods is very complete. Further, this type of effort is incredibly challenging, and the authors provide an good discussion about the uncertainties in the assumptions they made in their approach. Despite shortcomings in the data and models, this is a very good effort and will provide improvements to future model simulations. I only have minor suggestions and some editorial comments for the authors.

General Comments: I may have missed this, but I would assume that fire models that are described need to be forced with atmospheric inputs. It is unclear to me what forcing were applied in the simulations that produced the emission results. This should be made more clear somewhere in the paper. I am assuming that they were all driven by the same climatic drivers?
*The FireMIP models were all driven with the same climatic drivers as explained in the FireMIP protocol (http://www.imk-ifu.kit.edu/downloads/pai/FIREmip_protocol_web0.3.pdf). We have now rephrased this in the main text (P12 L01): "FireMIP used identical forcing datasets with prescribed meteorological forcing (1901-2013), global atmospheric $CO_2$ concentrations (1750-2013), lightning (1871-2010), land use change (1700-2013), and population density (1700-2013) (Rabin et al., 2016)."*

El Niño is obviously an important driver of fire activity and emissions, particularly in EQAS. This is not captured in the emission estimates before the 1970's. Is this a problem? Can the authors comment on this further?
*The title of the paper where this reconstruction is based on is "Human amplification of drought-induced biomass burning in Indonesia since 1960" (Field et al., 2009). The key message there is that it takes both humans and drought to get big fire events and the relation between ENSO and fire emissions becomes weaker when going back in time because there were*

*fewer humans aiming to convert the landscape. The reviewer is right that before the 1970's we will not capture those fires but the data indicates that emissions were low then anyway.*

Editorial Comments:
Page 3, line 16: Should it be "directly" and "indirectly"
*We changed this to directly and indirectly.*

Page 3, line 24: What other land surfaces? The previous sentence talks about deforestation fires. So, is this land surfaces other than forests?
*Yes, we changed this to: "For fires not associated with deforestation."*

Page 5, line 7: Change to "All of these" - *Done*
Page 5, line 14: Change "which" to "that" – *Done*

Page 6, lines 7-11: This is a very long sentence and could be broken up to read more clearly.
*We changed this to: "Based on CH₄ concentrations and its isotopic ratio, Ferretti et al. (2005) have hypothesized that this decrease of human-driven fires in the South American tropics was related to the arrival of Europeans and the introduction of diseases in the tropics. This would have decimated the population and lowered the number of human ignitions. However, decreased burning is evident in both the Americas and globally (Power et al., 2013), and thus is better explained by widespread cooling during the LIA."*

Page 6, line 18: The differences "over the past decade"? What is meant by this?
*We actually meant the past decades but have rephrased that part of the text (P6L17) to: "Although biomass burning reconstruction based on isotopic ratios of CO and those of CH₄ as well as those derived from charcoal records show similar features there are key differences. These are most pronounced for the past 50-100 years and could be the result of different lifetimes of CO (two months, providing more regional information) and CH₄ (about a decade, providing information on a global scale), but also because of the distribution of the charcoal datasets, which is denser in temperate regions than in the tropics."*

Page 7, lines 9 and 10: "data" are plural. Change to "provide" – *Done*

Figure 1: Shouldn't the satellite observations circle expand downward to local scales? *Yes, we have extended the Satellite observations circle more towards local scale and towards decadal scale (GFED4s is now available for 2 decades).*

Page 9, line 6: The reference for GFED4s should be provided. *We added the reference to GFED4s (van der Werf et al., 2017).*

Page 10, line 11: A comma should be used before the word "which" (here and throughout the paper). – *We have the checked the paper for "which" and added a comma if necessary.*

Page 11: The emission factors used in this are from Akagi et al. 2011. Did you include the emission factors from the updates to this dataset (from 2015). *Most of the emission factors are indeed from Akagi et al. (2011) but updates and other sources were used as well. This is detailed in van der Werf et al. (2017)and in the text we now refer to that paper (P11,L08): "As a final step, these carbon emission estimates are converted to trace gas and aerosol emissions using emission factors based mostly on the compilation of Akagi et al. (2011) but updates and other sources were used as well (van der Werf et al., 2017). An overview of the emission factors used in this study is given in Appendix C."*

Page 14, line 17: Change "which" to "that" – *Done*

Page 14, line 25: change the tense to be consistent ("are" should be "were") – *Done*

Page 16, line 3: How can you compare the visibility outputs to 1750 – 2000 when those data don't go back that far? This is unclear. – *The 1750-2015 reconstruction was based on GFED4s (1997-2015), visibility observations (1950-1996 for EQAS and 1973-1996 for ARCD), and the lowest decadal average from that time series for the pre-visibility time period. This time series was compared to HYDE population density. We changed this sentence to: "and extended visibility-based fire emissions using the lowest decadal average for the period before visibility observations became available"*

Page 17, line 7: all "of" our – *Done*

Page 18, line 22: Define IAV when first used. – *We defined IAV at P03L06.*

Page 18, line 24: Change "which" to "that" – *Done*

Page 28, line 2: Change to "there are very little data" – *Done*

Page 34, line 18-19: Current emissions? Does this mean the current emissions (2000? 2010?) in the CMIP5 estimates? – *Changed this sentence to: 'which was lower than their emission estimates in 2000'*

Page 34, line 21: Should "in" be "is"? – *No, in our opinion this sentence is grammatically correct.*

Page 36, line 14: This sentence is worded poorly and should be rewritten. Page 39, line 8: Should there be an ; or : after "emissions" – *We rephrased this to: 'We have assumed that fire emissions did happen at a much lower rate, either man-made or naturally. However, the relation between climate, humans and fires is complicated (Archibald, 2016).'*

*References:*

Akagi, S. K., Yokelson, R. J., Wiedinmyer, C., Alvarado, M. J., Reid, J. S., Karl, T., Crounse, J. D. and Wennberg, P. O.: Emission factors for open and domestic biomass burning for use in atmospheric models, Atmos. Chem. Phys., 11, 4039–4072, doi:10.5194/acp-11-4039-2011, 2011.

Archibald, S.: Managing the human component of fire regimes: lessons from Africa, Philos. Trans. R. Soc. B Biol. Sci., 371, 20150346, doi:10.1098/rstb.2015.0346, 2016.

Field, R. D., van der Werf, G. R. and Shen, S. S. P.: Human amplification of drought-induced biomass burning in Indonesia since 1960, Nat. Geosci., 2, 185–188, doi:10.1038/ngeo443, 2009.

Rabin, S. S., Melton, J. R., Lasslop, G., Bachelet, D., Forrest, M., Hantson, S., Li, F., Mangeon, S., Yue, C., Arora, V. K., Hickler, T., Kloster, S., Knorr, W., Nieradzik, L., Spessa, A., Folberth, G. A., Sheehan, T., Voulgarakis, A., Prentice, I. C., Sitch, S., Kaplan, J. O., Harrison, S. and Arneth, A.: The Fire Modeling Intercomparison Project (FireMIP), phase 1: Experimental and analytical protocols, Geosci. Model Dev. Discuss., 1–31, doi:10.5194/gmd-2016-237, 2016.

van der Werf, G. R., Randerson, J. T., Giglio, L., van Leeuwen, T. T., Chen, Y., Rogers, B. M., Mu, M., van Marle, M. J. E., Morton, D. C., Collatz, G. J., Yokelson, R. J. and Kasibhatla, P. S.: Global fire emissions estimates during 1997-2015, Earth Syst. Sci. Data Discuss., 1–43, doi:10.5194/essd-2016-62, 2017.

---

## Author Comment (AC4) · 15 May 2017

The paper aims at providing a historical reconstruction of fire emissions from 1750 onward, as the basis for the CMIP6 climate modeling objective. This paper then focusses on updating the 1850-2000 fire emissions proposed for the CMIP5 exercise. To reach this goal, the authors use the GFED4s emissions data as the baseline for 1997 to present period. The backward trend line for the Tropical forest is based on newly delivered papers reconstructing fire emissions since the 1960's based on visibility indices. The global charcoal database is used for boreal and temperate forest of the northern hemisphere where the network of sample is the most significant and from a panel of DGVMs runs for the 1750-present period for all the other areas.

The objectives are timely, and the effort in assembling the state-of-the-art modelling and charcoal communities deserves congratulations for proposing a synthesis. The strength of the paper in assembling 6 models, and readjusting the non-quantitative charcoal temporal variations to fit the final GFED4s time series, might also be however its main weakness. It is on one side a huge data assemblage, and on the other side a poorly investigated model intercomparison weakening the final message.

Despite being well and clearly described, some assumptions remain confusing and potentially misleading. The total absence of link and usage of the MIP5 reconstruction is also frustrating.
*As the reviewer mentioned, there has been substantial progress since the MIP5 reconstruction was produced by Lamarque et al. (2010). Our aim was to make best use of that new information which has most certainly led to improvements in several regions. At the same time, uncertainties remain substantial and we had to made a number of rather arbitrary choices which we have described as good as possible and we included sensitivity studies to estimate the impact of those choices.*
*Our estimates and the CMIP5 are actually more in line than the reviewer suggests. Both started using GFED but obviously using different version (2*

*versus 4s) and both went back in time using other datasets, which to some degree compare reasonably. By expanding the discussion on the newly presented dataset and MIP5 and further clarifying some of the assumptions we hope to have taken away most of the concerns of the reviewer.*

The main assumption of the paper is that "fire models can be used to estimate biomass burning emissions on a global scale"(P4l21-23), and this also on a long temporal scale.
In this sense, the paper contradicts itself when, in the end, comparing model's performances to charcoal data on selected regions, and concluding on poor relationships.
*The reviewer is right, and we may have not chosen our wording properly We have now rephrased this to: "fire models are also used to estimate biomass burning emissions on a global scale".*

In absence of any other data, we might understand however to rely on this data resource. I have listed below the questions I am concerned with, which would require major corrections and significant additional information. Unfortunately, I think this approach would really deserve a deeper FIREMIP result understanding before being used for this purpose.
*Ideally all the different models would have been evaluated before being used for an exercise like ours. This is actually done in FireMIP but it may take a number of years before those results become available. In the meantime, CMIP6 requires estimates that are based on the best knowledge currently available and that is what we have done. We believe science is an incremental process and we highlighted in several sections that uncertainties are substantial and that future reconstructions may be different, just like ours is different (but based on better science, especially in those areas where new constraints have emerged) than previous ones*

When going through the 3 main methodological tasks used for the reconstruction, I have the following questions:

1. Visibility: this interpolation based on two published papers linking visibility to GFED emissions for the period 1997-present and extending backward to 1960's in south east asia and Amazonia is really convincing, both in terms of temporal trend and interannual annual variability. In this sense, this is a significant update to the MIP5 reconstruction. It would be interesting though to have this comparison with MIP5 for all regions, to clearly understand the added value of this synthesis (as performed in figure 14).
*We appreciate the suggestions and have added regional comparison in the supplement to better inform the reader about differences between our estimates and previously used fire emissions estimates for CMIP. Reviewer 1 also raised this point. The figure is inserted below as well and we have added the following text to the discussion (P34L24):*

*"Although the global trends are relatively similar, on a regional scale differences between our estimates and the data used in CMIP5 are more substantial (See Figure D1, with regional comparisons between CMIP5 and CMIP6 estimates in Appendix D), with the largest differences in TENA-E, TENA-W, SHAF and SARC. In Africa, the continent of which half of all carbon emissions stem, we found that emissions were relatively flat while CMIP5 estimates increased over the past decades, at odds with recent findings that agricultural expansion lowers fire activity* (Andela and van der Werf, 2014). *The estimates and trends in EQAS, CEAS BONA-W, BONA-E are very similar, just as the estimates in ARCD, although in our estimates the increase there started a few decades later. While our estimates are for several regions driven by consistent data sources, these substantial discrepancies highlight once more that uncertainties are large"*.

[Figure]

*Figure D1 Regional carbon monoxide biomass burning emissions estimated by Lamarque et al. (2010) for CMIP5 and our results (CMIP6) on an annual and decadal time step.*

I have just a little concern that the Van Marle et al. (2017) paper used for this reconstruction analysed only a portion of the ARCD region showed in figure 2. Peru and Eastern Brazilian (fire-prone cerrado savannas) don't seem to be included in this temporal trend reconstructed from visibility. =>How did the authors deal with this other part of the ARCD region, still representing a significant surface?

*For this purpose we divided the GFED region 'Southern hemisphere South America' into a region dominated by deforestation (Arc of Deforestation, ARCD) and a region further south where cerrado fires, amongst other, occur. This was labeled South of the Arc of deforestation (SARC). Currently, fire-driven deforestation is the main source of carbon emissions in ARCD and compared to deforestation emissions other fire sources play a small role* (Morton et al., 2006, 2008; van der Werf et al., 2010)*. van Marle et al. (2017) showed that fire emissions were low up until the late seventies when deforestation practices started. Before 1973 (when visibility observations became available) we kept emissions at the lowest decadal value and assumed that this baseline corresponds to fires from cerrado burning.*

*We agree with the reviewer that these uncertainties could be described in more detail.*

*Therefore we added to the Methods P16L01:*
*"In ARCD deforestation emissions dominate the fire emissions, but additional emissions stem from cerrado burning. We assumed that fraction corresponds to our baseline emissions in the 1970s when deforestation was low and was kept constant before that period.'*

2. charcoal-based reconstruction The authors used the global charcoal database, providing a general trend in historical charcoal deposition in sediments from vegetation fires, with increasing time resolution allowing for decadal understanding of fire history. The authors selected the regions with a significant amount of data, which is a fair assumption. The main weakness of this dataset is the missing quantitative information so the authors had to rescale the Z-scores of the charcoal database to the emissions. The method is described in p17.

We get a little confused p17l9-10 with the sentence "the normalized charcoal signal (CCnorm) is the unitless charcoal influx Z-score on a decadal time step normalized per region and year". this is minor, but decadal and yearly time step sound confusing to the reader. That should be rephrased.

*We agree and rephrased p17 3-4 to in the revised manuscript to:*
*"The charcoal records were converted to unit less time series, with a range between -1 to 1, with a decadal time step using methods detailed in Power et al. (2010). The decadal time series was linearly interpolated to annual values and subsequently scaled to the output of the modelled data described under 2.2 following Eq. 2:"*

When looking at Power et al. 2010 and Marlon et al. 2016 papers, Z scores vary below 0 and above 1, so I guess these values have been reduced to the 0-1 interval. Is that correct? Maybe rephrase as we understand, as written, that Zscores are directly between 0 and 1 in the raw data. To rescale the Z score, the authors then assume that the maximum Z-score corresponds to the 75th percentile of FIREMIP models and the minimum z score to the 25th percentile in equation 2. This assumption is then thoroughly and properly discussed later.
*We rephrased p17 3-4 to: "The charcoal records were converted to unit less time series, with a range between -1 to 1, with a decadal time step using methods detailed in Power et al. (2010). The decadal time series was linearly interpolated to annual values and subsequently scaled to the output of the modelled data described under 2.2 following Eq. 2:"*

We wonder however in Equation 3 p17, why CCscaled is based on CCfireMIP of the year 2000 and not the mean 1997-2003 period as FIREMIPscaled (equation 1)?
*The charcoal data is available on a decadal time step, with $CC_{FireMIP}$ the scaled charcoal data to the $75^{th}$ and $25^{th}$ percentile of FireMIP data. To get the annual values we linearly interpolated the decadal charcoal observations. Taking the average of $CC_{FireMIP}$ over 1997:2003 would imply that we used annual observations and would furthermore not result in different outcomes than using the observed 2000 value.*

The output from this rescaling is finally a 10-year smooth average, without any interannual variability (as shown in figure 10 for example). Then why not using the FIREMIP interannual variability to produce this missing variability on the smooth charcoal trend? *The charcoal-based time series only provides values on a decadal time step. We decided only to use the FireMIP interannual variability in regions where also the trend is scaled based on FireMIP data. For regions where charcoal records were used we refrain from using the FireMIP interannual varability to clearly emphasize that the underlying charcoal trend is based on decadal data. Also, adding the interannual variability based on the FireMIP data would add additional uncertainty to the estimates in these regions, because this will require additional (rather arbitrary) assumptions on the size of the interannual variability.*

For the EU region, the charcoal database is used. Samples are distributed across Europe, while burned area is mostly located in the south on the mediterranean part. Are the charcoal sample locations weighted according to present observed burned area for example to give more weight to the Mediterranean? If not how biased could be the result? –
*Following earlier work* (Marlon et al., 2016) *we have not weighted the individual records but have strived to make the regions as small as possible. In North America the signals within the original GFED basis regions diverged and there we split those regions. That was not the case in Europe so there was no need to weigh them. In addition, Europe contributes 0.4% to total global emissions so*

*even a slightly different reconstruction would have negligible impact on the global signal*

For north America, The method is clearly described and discussed so that could be convincing. I still wonder here, however, why the authors did not use the forest fire statistics from US and Canada and reconstructions of burned areas going back in time for almost a century in these regions widely documented to rescale the minimum and maximum emissions? These data have been used in MIP5 and in my opinion would have greatly benefited here to strengthen the decision of this 25th and 75th percentile, and make a link to the previous version.
*We thank the reviewer for this suggestion and agree this might have been a valuable dataset. One of these datasets is the Canadian Fire Database (CNFDB), which nicely extends the satellite era decades back in time. While this dataset provides more spatial detail than our approach, users are also warned by CNFDB that the data in the CNFDB is not complete and not all fires have been mapped and data accuracy varies. The collection only includes data that has been contributed by agencies and completeness and quality may vary among agencies and between years. This makes the dataset less useful for our purpose than initially thought. Adding this dataset would need a thorough regional comparison with the charcoal time series and satellite-based emissions.*
*We added to the discussion P36L08: "Furthermore, an in-depth comparison between forest fire statistics from the US and Canada, for example the Canadian Fire Database (CNFDB, Stocks et al., 2002) and the charcoal time series may help constrain the uncertainty in boreal and temperate North America."*

3 DGVMs historical runs. In absence of any substantially reliable information, the authors decided to use the FIREMIP runs. The choice is clearly stated in the methods. It then covers a very significant portion of the globe (Africa, south America beside Amazonia, Asia, and Australia) and a large portion of the global burned area. Figure 3 could be rearranged proportionally to burned area, so that the reader clearly visualize that the global burned area reconstruction relies mostly (round 75% ) on models.

[Figure]

*Figure 3: Data sources used for each region. The pie chart represents the contribution of the modelled regions (purple), charcoal regions (green), and visibility-regions (grey) to the GFED totals over 1997 to 2015.*

I am not against this idea, but in turn, the reader is left a little disappointed and questioned as the paper doesn't analyse at all models assumptions and specificities. The authors give us the huge variability from the models (which is disappointing but actually in the range of uncertainties in climate model projections) and we don't really know what is climate-driven, human-driven and why each model has this trajectory. Analyzing all this would require one full (or even several) papers from this modelling group so they give us further information. and it's a huge task. I might understand the rush to provide CMIP6 data for burned area emissions, but this chapter leaves the reader very frustrated, if not suspicious on the reliability of these data for this purpose. I guess the authors would argue that it's still better than the empirical reconstruction from MIP5 and the linear trend used before 1900.when looking at figure 5 and the 1750-1900 trend, it's not obvious that the authors have achieved a fundamentally innovative trend compared to MIP5. –

*We agree with the reviewer that for those regions where the fire models were used our results may not be a clear improvement compared to earlier work. This is now mentioned more clearly in the text by adding in the discussion P36 L08: "Future model comparisons pinpointing the reasons why models behave differently would help constrain this uncertainty."*

*Furthermore we compared the MIP5 and MIP6 results on a regional scale (see comment before) and added to the Discussion P34L24: "Although the global trends are relatively similar, on a regional scale differences between our estimates and the data used in CMIP5 are more substantial (See Figure D1, with regional comparisons between CMIP5 and CMIP6 estimates in Appendix D), with the largest differences in TENA-E, TENA-W, SHAF and SARC. In Africa, the continent of which half of all carbon emissions stem, we found that emissions were relatively flat while CMIP5 estimates increased over the past decades, at odds with recent findings that agricultural expansion lowers fire activity (Andela and van der Werf, 2014). The estimates and trends in EQAS, CEAS BONA-W, BONA-E are very similar, just as the estimates in ARCD, although in our estimates the increase there started a few decades later. While our estimates are for several regions driven by consistent data sources, these substantial discrepancies highlight once more that uncertainties are large".*

*We also provided sensitivity analyses to the number of models included in the trend derivation. For Africa (~50% of total fire carbon emissions) we do feel we have improved as the multi-model mean is more in line with recent findings about the role of agriculture in suppressing fires (Andela and van der Werf, 2014) than MIP5 where biomass burning there increased over time (Figure D1).*

When going into details on this chapter, I have the following questions:

- P12 l2: FIRE MIP runs DGVMs from 1700 to 2013. GFED from 1997 to 2015. The overlapping period is 1997-2013. Why using 1997-2003 further on (line 5) as an overlapping period?
*We used the 1997-2003 period as benchmark to scale the modeled data to and we prefer to use GFED data from 1997 onwards. To use the GFED data as benchmark we decided to scale the 2000 value of the modeled data, because there is still some overlap with the GFED time period. We do understand this decision is rather arbitrary, but using the whole overlapping time period with the models would result in a mismatch when stitching the modeled data to GFED, because trends in the time period over 1997-2013 occur. We also rephrased P12 L12-14 to: "We used the average over 1997 to 2003 when combining the various data streams to minimize the impact of interannual variability in the GFED time series, which could result in a mismatch when stitching the FireMIP emissions to the GFED data."*

timing of interannual variability: I was expecting that, if the trend is not overwhelmingly different from the flat trend of MIP5, we would get the actual interannual variability in time and amplitude from this approach. We also get a little disappointed as all experiments used repeated 1901-1920 forcings from the beginning of the simulation (1750) to 1900. In this sense, figure 5 is misleading and should better be presented as a moving window decadal values with uncertainties (SE or coeff of variation), as the variability is not timely.

We agree with the reviewer that the 20-year cyclic meteorological forcing should be mentioned more clear in the text. Therefore we added to the discussion P32L02: "Meteorological forcing data was only available for the year 1900 onwards. The interannual variability before 1900 stems from a 20-year repetitive cycle in meteorological forcing (1900-1919)."

Although the IAV in the FIreMIP data is based on a 20-year repetitive cycle for the meteorological forcing before the year 1900, other forcing data such as land-use, population density and $CO_2$ concentrations were available before 1900.This provides information based on the model output we would like to keep included.

We investigated the effect of taking the 20-year running mean over every modeled time series on the regional and global results. On a global scale the differences are marginal with 0.2%. On a regional scale the differences go up to 7% in NHSA, although this region contributes only 1.4% to the global totals. We prefer to keep our results including IAV, however as described above we will describe the 20-year repetitive cycle more clear in the text.

*Table R1 – Carbon emissions based on using the models including interannual variability (IAV), models with a 20-year moving window and the difference relative to the current estimates (IAV-based).*

| | | Average emissions (incl. IAV) (Tg C year-1) | Emissions (20-year moving window) (Tg C year-1) | Relative difference (%) |
|---|---|---|---|---|
| BONA-W | Boreal North America – West | 41.1 | 39.7 | 3.2 |
| BONA-E | Boreal North America – East | 12.5 | 12.1 | 3.2 |
| TENA-W | Temperate North America - West | 8.4 | 8.4 | 0.9 |
| TENA-E | Temperate North America – East | 14.1 | 13.7 | 2.9 |
| CEAM | Central America | 44.5 | 44.0 | 1.2 |
| NHSA | Northern Hemisphere South America | 26.4 | 28.4 | -7.7 |
| ARCD | Arc of Deforestation | 57.7 | 57.7 | 0 |
| EURO | Europe | 7.0 | 7.1 | -1.2 |
| MIDE | Middle East | 3.1 | 3.1 | 0.6 |
| NHAF | Northern Hemisphere Africa | 475.4 | 475.4 | 0.01 |
| SHAF | Southern Hemisphere Africa | 623.3 | 615.8 | 1.2 |
| BOAS | Boreal Asia | 101.3 | 104.7 | -3.4 |
| CEAS | Central Asia | 78.2 | 80.6 | -3.1 |
| SEAS | South-East Asia | 207.3 | 207.1 | 0.2 |
| EQAS | Equatorial Asia | 47.3 | 47.3 | 0 |
| AUST | Australia | 97.4 | 97.2 | 0.2 |
| SARC | South of Arc of Deforestation | 51.3 | 50.8 | 0.9 |
| GLOBE | Sum of all regions | 1896.4 | 1893.0 | 0.17 |

Also why minimizing interannual variability ( P12 L12-L14) on purpose? The authors in additions discuss about the increasing interannual variability but the trend of this variability in figure 5 is all fake.

*As described before after 1900 the IAV in the FireMIP data is based on meteorological forcing data. Only before 1900 the IAV in the FireMIP data is based on a 20-year repetitive cycle for the meteorological forcing, although other forcing data such as land-use, population density and $CO_2$ concentrations were available before 1900.*

*The sentence written on P12 L12-14 is to explain how we stitched the modeled data to the GFED data. The GFED data has interannual variability and just matching the modeled data to the 2000 value of GFED observations would result in a mismatch, because the models don't exhibit the same inter annual variability. Therefore we used an average over 1997-2003. We feel this comment was the result of a misunderstanding so we rephrased the section to be more precise: "We used the average over 1997 to 2003 when combining the various data streams to minimize the impact of interannual variability in the GFED time series, which could result in a mismatch when stitching the FireMIP emissions to the GFED data."*

This should not be taken for granted as: a) considering the mean when emission simulations are not timely in phase for each model (figure 7 for example) intrinsically reduces the interannual variability (lower than each model's interannual variability) , *All models used identical meteorological forcings as such the emission simulations are timely in phase for each model.*

b) the charcoal time serie is flat (discussed above). Why do the authors provide this 'fake' interannual variability ? is that a request from the CMIP6? It would be worth, in the introduction for exemple, to present the CMIP6 'wish list' to better understand the choices perfomed in this reconstruction. *This comment links to an earlier comment raised by the reviewer. We investigated the effect of taking the running mean over every modeled time series on the regional and global results (Table R1), which shows that the differences are marginal.*

We are also questioned that the authors used the 25th and 75th percentiles for charcoal reconstruction using FIREMIP models, so that "outliers did not influence the scaled regional charcoal signal" (P15L15). We then wonder why this was not also done for equation 1. *In Equation 1 we did not scale the modeled data, because they have their own upper and lower limit corresponding to emissions. Charcoal needed the scaling in order to get values corresponding to the Z-scores and the models needed the scaling to match the GFED data. Furthermore we took the median for the modeled regions, which in turn reduces the effect of outliers.*

In conclusion for this modelling chapter, if we can knowledge the effort of the authors to assemble all this information, the conclusions seem way too overrated and we miss a lot of the understanding of this model intercomparison to fully appreciate the synthesis. The interannual variability is an important point that is completely misrepresented in the final results and misleading for the readers.
*We agree with the reviewer this should be highlighted more. Therefore we added to the discussion P32L02: "The interannual variability before 1973 stems from a 20-year repetitive cycle in climate forcing used in the models."*
*We highlighted the need for future model studies by adding in the discussion P36 L08: "Future model comparisons pinpointing the reasons why models behave differently would help constrain this uncertainty."*

Discussion: The discussion is interesting and actually provides more interesting information than the results themselves. However, it also highlights the weakness of the results.
P32 l1: we wonder if the visual trend is actual or driven by the "fake" interannual variability. Statistical time series analysis could reinforce this sentence, but with a wrong interannual variability they will be also biased.
*We have estimated regional and global carbon emissions based on the data presented and the datasets smoothened (see Table D1) showing that the difference is marginal.*

P32 l13-14: "after which emissions stabilized, probably as a result of increasing CO2 concentrations and changes in population density as input parameters" This sentence clearly illustrates my comments on the poor analysis of the models functioning. It is very difficult here to understand and have an opinion based on the information provided in the paper (neither by reading hantson et al 2016 and Rabin et al describing the models): why increasing CO2 would stabilize fire emission?

*We agree with the reviewer and have eliminated the speculative part: "The multi-model median indicated that Southern Hemisphere Africa (SHAF) had an increasing trend from 1750 until ~1950, after which emissions stabilized."*

For SAH, different trends are observed in models. . .but all are driven by population (at least ORCHIDEE and LPJ GUESS SPITFIRE are coupled with the same SPTIFIRE but with the most opposite trends...). A full model output analysis would be worth being published before this paper, to strengthen the message.

*All models performed differently although the input datasets were similar. The FireMIP community is currently working on detailed intercomparison analyses and benchmarking practices. Although the exact pattern in models is unclear the models provide currently the most continuous datasets and are the sources to rely on especially in regions where little is known about fire history. We do make the assumption that the median is most representative, but until detailed model intercomparison analyses are done we don't know which model performs where best. We agree with the reviewer that this is a limitation of our results. We highlighted this comment more by adding in the discussion P36 L8:*
*"Future model comparisons pinpointing the reasons why models behave differently would help constrain this uncertainty."*

Figure 13 p 33: Using the Andela and van der Werf (2014) hypothesis seems to be a fair option to reconstruct fire history actually for Africa. That's a nice result. Why not choosing this trend the same way the authors did with charcoal? This would completely reverse the global increasing trend obtained from the FIREMIP into a decreasing trend, and would fit the charcoal Tierney (2010) trend. That sounds convincing.

*We agree that the Andela and van der Werf (2014) method provides insight into fire behavior in Africa. However their method is solely based on the satellite era. Patterns and causes of fires in Africa might have changed over the century. Our method yields a somewhat different trend. However we do agree this highlights the uncertainty of the global trend, which is for a large part based on the African signal. Therefore we added the following sentence to the discussion P32L25:*
*"Future research into the drivers of African fires and how these have changed over time could help would improve our estimates."*

How is cropland area introduced in DGVMs? If not included, there is no reason to value the model hypothesis rather than the Andela paper. This paragraph is

again both exciting as the authors seem to have found a smart proxy fitting the charcoal but they don't use it, but also disappointing as it weakens the model's approach, that we are not able to fully appreciate due to a lack of deep analysis. *In previous versions of the dataset we indeed used agriculture as a proxy. After discussions with the fire modelers this was changed mostly because we felt it was inappropriate that over 250 years of fire emissions were a function of only 1 parameter, given that over the same time frame several other crucial parameters (grazing, $CO_2$ fertilization, other land cover changes) have changed. We totally agree that no model models this perfectly but at least most factors are accounted for including changes in cropland area. Again, this is a subjective decision and the reviewer is right in questioning this, but any other choice would have been subjective as well. By highlighting three different approaches (fire model mean, agriculture as a proxy, and charcoal) to estimate emissions from Africa we have highlighted the uncertainty in this.*

The final discussion chapter on the comparison with MIP5 is welcome (at last!). Too bad it's partial and only focused on few areas. A final comparison on the MIP5 and MIP6 would be also interesting. . . as the MIP6 seems to be flat before 1900, and it sounds like it would be very similar to MIP5 in the end. *We compared the MIP5 and MIP6 results on a regional scale (see comment before) and added to the Discussion P34L24: "Although the global trends are relatively similar, on a regional scale differences between our estimates and the data used in CMIP5 are more substantial (See Figure D1, with regional comparisons between CMIP5 and CMIP6 estimates in Appendix D), with the largest differences in TENA-E, TENA-W, SHAF and SARC. In Africa, the continent of which half of all carbon emissions stem, we found that emissions were relatively flat while CMIP5 estimates increased over the past decades, at odds with recent findings that agricultural expansion lowers fire activity* (Andela and van der Werf, 2014). *The estimates and trends in EQAS, CEAS BONA-W, BONA-E are very similar, just as the estimates in ARCD, although in our estimates the increase there started a few decades later. While our estimates are for several regions driven by consistent data sources, these substantial discrepancies highlight once more that uncertainties are large".*

Some few minor additional comments:
P3L8 : the varying constraint hypothesis from krawchuk and moritz 2011 would be a better reference in addition or replacement of van der werf 2008. *We thank the reviewer for the suggestion and added the paper as reference.*

P4l21-23: this is a critical assumption that "fire models can be used to estimate biomass burning emissions on a global scale" on a historical point of view. . . maybe review some recent papers trying to compare historical trends (Yue et al., Kloster et al., Yan et al.). *We rephrased this sentence (see earlier comments) to:' "fire models are also used to estimate biomass burning emissions on a global scale"*

P18 l 22: IAV? Does it mean interannual variability? *We defined IAV where it was first introduced at P03L06.*

P38: figure 14: just wondering if charcoal Z-scores should be rescaled to the 50 year average of burned area from Mouillot & field and C emissions from your study to better rescale the temporal trend, instead of year 2000*. In Figure 14 the three datasets (Charcoal Z-scores, Mouillot and Field and Our estimates) were normalized and scaled to their 2000-values. The three datasets are for these regions independent of each other and this way it is possible to compare the trends as objective as possible.*

*References*
Andela, N. and van der Werf, G. R.: Recent trends in African fires driven by cropland expansion and El Niño to La Niña transition, Nat. Clim. Chang., 1–5, doi:10.1038/nclimate2313, 2014.
Lamarque, J.-F., Bond, T. C., Eyring, V., Granier, C., Heil, A., Klimont, Z., Lee, D., Liousse, C., Mieville, A., Owen, B., Schultz, M. G., Shindell, D., Smith, S. J., Stehfest, E., Van Aardenne, J., Cooper, O. R., Kainuma, M., Mahowald, N., McConnell, J. R., Naik, V., Riahi, K. and van Vuuren, D. P.: Historical (1850–2000) gridded anthropogenic and biomass burning emissions of reactive gases and aerosols: methodology and application, Atmos. Chem. Phys., 10, 7017–7039, doi:10.5194/acp-10-7017-2010, 2010.
van Marle, M. J. E., Field, R. D., van der Werf, G. R., Estrada de Wagt, I. A., Houghton, R. A., Rizzo, L. V., Artaxo, P. and Tsigaridis, K.: Fire and deforestation dynamics in Amazonia (1973-2014), Global Biogeochem. Cycles, 31, 24–38, doi:10.1002/2016GB005445, 2017.
Marlon, J. R., Kelly, R., Daniau, A.-L., Vannière, B., Power, M. J., Bartlein, P., Higuera, P., Blarquez, O., Brewer, S., Brücher, T., Feurdean, A., Romera, G. G., Iglesias, V., Maezumi, S. Y., Magi, B., Courtney Mustaphi, C. J. and Zhihai, T.: Reconstructions of biomass burning from sediment-charcoal records to improve data–model comparisons, Biogeosciences, 13, 3225–3244, doi:10.5194/bg-13-3225-2016, 2016.
Morton, D. C., DeFries, R. S., Randerson, J. T., Giglio, L., Schroeder, W. and van der Werf, G. R.: Agricultural intensification increases deforestation fire activity in Amazonia, Glob. Chang. Biol., 14, 2262–2275, doi:10.1111/j.1365-2486.2008.01652.x, 2008.
Morton, D. C., DeFries, R. S., Shimabukuro, Y. E., Anderson, L. O., Arai, E., del Bon Espirito-Santo, F., Freitas, R. and Morisette, J.: Cropland expansion changes deforestation dynamics in the southern Brazilian Amazon, Proc. Natl. Acad. Sci., 103, 14637–14641, doi:10.1073/pnas.0606377103, 2006.
Stocks, B. J., Mason, J. A., Todd, J. B., Bosch, E. M., Wotton, B. M., Amiro, B. D., Flannigan, M. D., Hirsch, K. G., Logan, K. A., Martell, D. L. and Skinner, W. R.: Large forest fires in Canada, 1959–1997, J. Geophys. Res., 108, 8149, doi:10.1029/2001JD000484, 2002.
van der Werf, G. R., Randerson, J. T., Giglio, L., Collatz, G. J., Mu, M., Kasibhatla, P. S., Morton, D. C., DeFries, R. S., Jin, Y. and van Leeuwen, T. T.:

Global fire emissions and the contribution of deforestation, savanna, forest, agricultural, and peat fires (1997–2009), Atmos. Chem. Phys., 10, 11707–11735, doi:10.5194/acp-10-11707-2010, 2010.

---

## Author Response (AR1)

A. Kerkweg - kerkweg@uni-bonn.de Received and published: 24 February 2017

Dear authors,

In agreement with the CMIP6 panel members, the Executive editors of GMD would like to establish a common naming convention for the titles of the CMIP6 experiment description papers.

The title of CMIP6 papers should include both the acronym of the MIP, and CMIP6, so that it is clear this is a CMIP6-Endorsed MIP.

Good formats for the title include:

'XYZMIP contribution to CMIP6: Name of project' or
'Name of Project (XYZMIP) contribution to CMIP6'
If you want to include a more descriptive title, the format could be along the lines of, 'XYZMIP contribution to CMIP6: Name of project - descriptive title'

or 'Name of Project (XYZMIP) contribution to CMIP6: descriptive title.'

When you revise your manuscript, please correct the title of your manuscript accordingly.

Additionally, we strongly recommend to add a version number to the MIP description. The reason for the version numbers is so that the MIP protocol can be up- dated later, normally in a second short paper outlining the changes. See, for example: http://www.geosci-model-dev.net/special\_issue11.html,

Yours, Astrid Kerkweg

**Dear Astrid Kerkweg,**

Thank you for your comment. Based on the naming conventions for the CMIP6 experimental descript papers we have changed the title of our manuscript as follows: "Historic global biomass burning emissions for CMIP6 (BB4CMIP) based on merging satellite observations with proxies and fire models (1750-2015)" Kind regards, Margreet van Marle Interactive comment on "Historic global biomass burning emissions based on merging satellite observations with proxies and fire models (1750–2015)" by Margreet J. E. van Marle et al. Anonymous Referee #1

Received and published: 9 March 2017

This paper provides a description of the biomass burning emissions that are provided for the upcoming CMIP6 simulations. The authors have done an excellent job of providing in-depth description of the methodologies used to generate the emissions. This was a gargantuan task and the authors should be congratulated to achieving this. I have a small number of minor comments below.

My main complaint is that the emissions are showing fairly significantly different trends from the CMIP5 dataset and it would have been very useful if some model simulations (or at least estimates of radiative forcing) had been performed to understand the consequences of these different trends. I understand that this probably beyond the scope of this paper, but it is still a shortcoming worth mentioning.

This dataset will be used in the CMIP model simulations and presenting the results from that exercise is indeed beyond the scope of this paper. We have, however, added a more general statement to the conclusions: "Our results point towards less variability over time than the fire emissions used in CMIP5 and a smaller difference between pre-industrial and present fire emissions, lowering the impact on changes in atmospheric composition and potentially lowering overall radiative forcing".

**Minor comments**

Page 2, line 23: CMIP is not part of IPCC. It is part of WCRP (see Eyring et al., GMD, 2016)

We will change this to: 'Will be used in the CMIP6 simulations.'

Page 11, line 13: how large was the scaling when applied? Might be good to mention the scaling algorithm (Eq. 1) at this point. Since 1997 was such a large emission year, has its role been evaluated?

The scaling was done using Eq. 1. To be more specific we have added a reference to this in the sentence the reviewer mentioned: "were scaled (Eq. 1) to GFED4s." Scaling was based on the average of 6 years (1997-2003) as representative for the 2000 value of the models. An average was used to smoothen the effect of regional differences and the effect of interannual variability over the first years. The reason that 1997 was such a high fire year stemmed mostly from one region (Equatorial Asia where the El Niño induced drought that year led to record high emissions mostly from peat burning). For this region Eq. 1 is not used but is reconstructed using visibility observations. We have added to P12 L09: "where FireMIPscaled(reg,yr,mod) is the scaled regional

model output on an annual time step and FireMIP1997:2003(reg,mod) is the average regional estimate for 1997-2003. While this 7-year time period included the highest fire year, 1997, fire emissions in that year stem mostly for peat fires in Equatorial Asia for which Eq. 1 is not used to reconstruct fire emissions (See Sect 2.3)."

**Section 2.3: it seems that it would be useful to have more details on the methods used to extract emissions from visibility data? How does this work in anthropogenically polluted areas?**

We refer the reader to the papers on which this scaling is based (Field et al., 2009; van Marle et al., 2017) for more details in the methods. We agree with the reviewer that other sources impact visibility but we found these were of much smaller amplitude and do not influence the seasonal pattern used in our approach. Specifically, in both EQAS and ARCD visibility observations in low fire years at the end of our study period returned to similar levels as low fire years early in the study period indicating that other sources were of secondary importance.

Page 17, lines 26-27: any suggestions on how models could integrate that recommendation? 'When fire modules are embedded in climate models they may be in a better position to include some spatial and temporal variability based on simulated weather.'

Climate models that include fire models can calculate emissions directly, which may better capture spatial and temporal variability due to, for example, modeled weather patterns. We therefore inserted that sentence. There is no need for integration, because those models will not use our emissions estimates. To avoid confusion, we have rephrased the sentence to: "Those climate models that already have fire modules and calculate emissions directly may be in a better position to include some spatial and temporal variability based on simulated weather."

Page 18: change link to emission factors to an actual description in supplement. Web link will break over time

We added a table with the emission factors used for the different species in the appendix and refer to this in the text.

**Comparison with CMIP5: it would be greatly helpful if regional comparisons were also shown, maybe simply in the supplemental material**

We appreciate the suggestions and have added regional comparison in the supplement to better inform the reader about differences between our estimates and previously used fire emissions estimates for CMIP. The figure is inserted below as well and we have added the following text to the discussion (P34L24): "Although the global trends are relatively similar, on a regional scale differences between our estimates and the data used in CMIP5 are more substantial (See Figure D1, with regional comparisons between CMIP5 and CMIP6 estimates in Appendix D), with the largest differences in TENA-E, TENA-W, SHAF and SARC.

In Africa, the continent of which half of all carbon emissions stem, we found that emissions were relatively flat while CMIP5 estimates increased over the past decades, at odds with recent findings that agricultural expansion lowers fire activity (Andela and van der Werf, 2014). The estimates and trends in EQAS, CEAS BONA-W, BONA-E are very similar, just as the estimates in ARCD, although in our estimates the increase there started a few decades later. While our estimates are for several regions driven by consistent data sources, these substantial discrepancies highlight once more that uncertainties are large".